# ALPHAZERO-BASED PROOF COST NETWORK TO AID GAME SOLVING

**Ti-Rong Wu**[1*]**, Chung-Chin Shih**[1,2]**, Ting Han Wei**[3*]**, Meng-Yu Tsai**[1]**, Wei-Yuan Hsu**[1]**, I-Chen Wu**[1,2]

[1]Department of Computer Science, National Yang Ming Chiao Tung University, Hsinchu, Taiwan
[2]Research Center for Information Technology Innovation, Academia Sinica, Taiwan
[3]Department of Computing Science, University of Alberta, Edmonton, Canada
{kds285, rockmanray}@aigames.nctu.edu.tw, tinghan@ualberta.ca
{atuno, in13_13}@aigames.nctu.edu.tw, icwu@cs.nctu.edu.tw

## ABSTRACT

The AlphaZero algorithm learns and plays games without hand-crafted expert knowledge. However, since its objective is to play well, we hypothesize that a better objective can be defined for the related but separate task of solving games. This paper proposes a novel approach to solving problems by modifying the training target of the AlphaZero algorithm, such that it prioritizes solving the game quickly, rather than winning. We train a Proof Cost Network (PCN), where *proof cost* is a heuristic that estimates the amount of work required to solve problems. This matches the general concept of the so-called *proof number* from proof number search, which has been shown to be well-suited for game solving. We propose two specific training targets. The first finds the shortest path to a solution, while the second estimates the proof cost. We conduct experiments on solving 15x15 Gomoku and 9x9 Killall-Go problems with both MCTS-based and focused depth-first proof number search solvers. Comparisons between using AlphaZero networks and PCN as heuristics show that PCN can solve more problems.

## 1 INTRODUCTION

There are two main goals in the pursuit of strong game-playing agents. The first is to push the boundaries of artificial intelligence since games can be seen as simplified models of the real world. The second involves finding *game-theoretic values*, or outcomes given optimal play, for various games (van den Herik et al., 2002). These two closely related yet separate goals are commonly referred to as *playing* and *solving*[1] games, respectively. For playing, researchers have found success in building super-human level game-playing agents for many games, including Go (Silver et al., 2017; 2016), Chess, Shogi (Silver et al., 2018), and Atari games (Schrittwieser et al., 2020; Mnih et al., 2015). Achievements for solving include checkers (Schaeffer et al., 2007), Hex up to board sizes of 9x9 (Pawlewicz & Hayward, 2013; Henderson et al., 2009), Go for board sizes up to 5x6 (van der Werf & Winands, 2009), 15x15 Gomoku (Allis, 1994), among others.

Previously, researchers used hand-crafted heuristics with game-specific knowledge, combined with search methods such as Monte Carlo tree search (MCTS) (Winands et al., 2008), proof number search (PNS) (Allis et al., 1994), depth-first proof number search (DFPN) (Nagai, 2002; 1999), and threat-space search (Allis et al., 1993), to solve problems by massively pruning away unnecessary branches (Schaeffer et al., 2007; van der Werf et al., 2003; Allis et al., 1996; Allis, 1994). With the great success of AlphaGo and AlphaZero, there have been attempts to combine neural networks with previous solvers to further improve game solving. For example, Gao et al. (2017) incorporated an AlphaGo-like neural network into a state-of-the-art Hex solver to solve 8x8 Hex openings more quickly, which demonstrated that neural networks can have a positive impact when combined with previous techniques and heuristics. Meanwhile, McAleer et al. (2018) also trained an AlphaZero-

---

[*]These authors contributed equally.

[1]A solution to a game is an exhaustive strategy that guarantees the game-theoretic value against all opposing actions from the game's initial position (Allis, 1994).

like agent to solve the Rubik's Cube. In addition, game solving techniques were also shown to be viable for non-game applications (Kishimoto et al., 2019; Segler et al., 2018).

However, a limitation of using these techniques is that the networks trained with the objective of strong play may not be well-suited for obtaining game-theoretic values. Since the goal of a strong player is to play moves that maximize win rates (or the highest confidence for winning), as a heuristic it often does not choose moves that result in the fewest nodes in the search tree when solving. As an example, Agostinelli et al. (2019) pointed out that AlphaZero-like systems tend to solve problems with longer solutions. For games with a singular end state, e.g. Rubik's cube and n-puzzles, they proposed an approach that trains a cost-to-go value function that approximates the cost of finding the shortest-path solution, which can be viewed as a kind of prioritized-sweeping (Moore & Atkeson, 1993) algorithm, starting from the end state (goal). However, singular end states are often only exploitable for puzzles. Indeed, in two-player zero-sum games such as Go, Hex, Gomoku, Connect6, and many others, the game may end in too many configurations to enumerate practically. This paper presents a novel approach to solving these problems, while still tending to choose moves that lead to the fastest solutions.

In our approach, we propose a new heuristic, referred to as the *proof cost*. We give two concrete examples of how proof cost can be designed. The first acts as a "shortest path" heuristic to a proof, while the second predicts solution tree sizes. Then, we use AlphaZero training to approximate this heuristic. The resulting network is referred to as the *Proof Cost Network* (*PCN*), which can be used as a heuristic with any AND/OR tree search algorithm to improve solving efficiency.

Experiments were conducted on 15x15 Gomoku and 9x9 Killall-Go problems. The results show that PCN outperforms AlphaZero when used as a heuristic with MCTS-based and Focused DFPN (Gao et al., 2017; Henderson et al., 2009) solvers, which are both commonly used to solve games.

## 2 BACKGROUND

### 2.1 THE GAME-THEORETIC VALUE AND PROOF TREES

For a one or two-player game, the *game-theoretic value* of a game state is the best outcome that can be obtained by all players if they play optimally. In this paper, we focus on two-player games, and simplify the outcome to only include win/loss; draws are considered losses for both players[2]. A game state is said to be *solved* when the game-theoretic value for the state is obtained, and *proved/disproved* if the player to play for the state has a winning/losing outcome. In a proof, the winning player needs to ensure a winning response to all possible actions by the losing player. A *proof tree* (Pijls & de Bruin, 2001; Stockman, 1979) is an AND-OR search tree representing a strategy of answering actions such that (a) all terminal nodes are wins, (b) the winning player (the OR-player) contains at least one winning action, (c) all losing player (the AND-player) actions are enumerated, and the OR-player needs to have winning responses to each AND-player action. A *disproof tree* is the dual case of the above (with win/loss and AND/OR reversed); a *solution tree* is a general term used to describe both proofs and disproofs.

### 2.2 MONTE CARLO TREE SEARCH

Monte Carlo tree search (MCTS) (Coulom, 2006; Kocsis & Szepesvári, 2006) is a heuristic best-first search algorithm that iteratively repeats the following four phases: selection, expansion, evaluation, and backpropagation. In the selection phase, starting from the root node, the algorithm chooses a child node according to a specified selection criterion until a leaf node is reached. Given a state $s$ during selection, AlphaZero selects an action $a^*$ by the PUCT equation (Silver et al., 2018; Rosin, 2011):

$$a^* = \arg\max_a \left\{ Q(s,a) + c_{PUCT} \times P(s,a) \times \frac{\sqrt{\sum_b N(s,b)}}{1 + N(s,a)} \right\}, \quad (1)$$

where $Q(s,a) = W(s,a)/N(s,a)$ is the estimated mean win rate, $N(s,a)$ is the number of times $a$ is selected at $s$, $W(s,a)$ is the number of wins, $c_{PUCT}$ is a weight for exploration, and $P(s,a)$ is a

---

[2]We can solve the same problem twice, once for each player; if both outcomes are losses, then we know the outcome is a draw.

prior knowledge heuristic. The above selected leaf then expands all children which are added to the tree. The leaf node is then evaluated. In AlphaZero, a two-head network is trained such that it outputs a policy $p(s, a)$, and a value $v(s)$. The policy is a distribution that estimates how likely each child of $s$ will be visited during selection; $p(s, a)$ can then be used as $P(s, a)$ in eq. 1. The scalar $v(s)$ is an estimate for the game-theoretic value of $s$. During backpropagation we move back up the selection sequence and $Q(s_i, a_i)$ is updated for all ancestors $s_i$. While originally designed for playing, MCTS can also be used to solve games with AND-OR Boolean backpropagation. MCTS has been used to solve games and problems for Lines of Action (Winands et al., 2010; 2008), Connect4, Seki in the game of Go (Cazenave & Saffidine, 2010), and Connect6 (Wei et al., 2015).

## 2.3 PROOF NUMBER SEARCH

Proof number search (PNS) (Allis et al., 1994) is a best-first search algorithm that was designed specifically for solving games. Each node is associated with two numbers: the *proof number* (*PN*) and the *disproof number* (*DN*), which are heuristics that represent the minimum required number of leaf nodes that need to be expanded to prove and disprove the node respectively. PNS largely follows the same four phases as in MCTS. What is called the *most-proving node* (*MPN*) is selected during the selection phase. Specifically, the child with the lowest PN at each OR node, or the lowest DN at each AND node is selected; the resulting leaf node is then the MPN that needs to be evaluated. Following the definition of the PN/DN, PNS therefore builds the solution tree by expanding and evaluating the least number of nodes. There are many PNS variants, one of which is the so-called depth-first proof number search (DFPN) (Nagai, 2002; 1999), which addresses the large memory requirements for large searches.

PNS was originally designed such that all newly expanded nodes initialize PN/DN values to 1/1. To improve the heuristic, domain knowledge can be used to initialize PN/DN values differently, which speeds up the search for a solution (Winands & Schadd, 2010; Wu et al., 2010; Saito et al., 2006; Kishimoto & Müller, 2005; Allis, 1994). In addition, Henderson (2010) proposed Focused DFPN (FDFPN), which uses heuristics to direct the search toward promising solutions with a technique that is often referred to as progressive widening. Specifically, progressive widening involves prioritizing search on only a portion of all possible moves. The branching factor $b$ at an internal search node is dynamically adjusted by $b = b_{base} + \lceil \mu \times |b_{livechildren}| \rceil$, where $b_{base}$ is set to 1, $b_{livechildren}$ is the number of all yet unsolved children of the node, and $\mu$ is a weighting constant. With an accurate heuristic, FDFPN can solve problems while searching fewer nodes than unmodified DFPN. Further, Gao et al. (2017) proposed FDFPN-CNN by incorporating the policy and value outputs from an AlphaGo-based agent as heuristics to solve 8x8 Hex. Namely, the policy was used for move ordering and the value was used to control the branching factor by $b = b_{base} + \lceil f(s) \times |b_{livechildren}| \rceil$ and $f(s) = \min\{\mu, 1 + v(s)\}$.

## 2.4 THE ALPHAZERO ALGORITHM

AlphaZero (Silver et al., 2018) is a general reinforcement learning algorithm that can achieve super-human level playing strength for games without requiring any expert domain knowledge. The training routine consists of two phases: self-play and optimization. In a nutshell, a two-headed neural network that outputs a policy distribution $p$ and a value scalar $v$ is periodically optimized with data that is collected via self-play. Self-play games are generated by running a set number of MCTS simulations with the most recently optimized neural network. In the optimization phase, random positions are sampled from the latest self-play games, where the network is optimized by the following loss function:

$$L = (z - v)^2 - \pi^{\mathrm{T}} \log p + c\|\theta\|^2, \tag{2}$$

where $z$ is the ground truth outcome of the game for that position, $\pi$ is the MCTS search distribution, $c$ is an L2 regularization weight, and $\theta$ represents the network parameters.

## 3 OUR APPROACH

### 3.1 ESTIMATING PROOF TREE SIZES

We first examine the size of a proof tree, i.e. the number of nodes that need to be searched to obtain a proof for a *player of interest* from a given state. Since we do not know the game-theoretic outcome of the game before finding a solution, either player can be designated the player of interest. Given limited time and resources, it is almost always preferable to prove a given state by traversing as small a search tree as possible, where the ideal case is called a *minimum proof tree*. A minimum proof tree, say rooted at a node representing state $s$, is defined as a proof tree containing the least possible number of tree nodes, denoted by $n(s)$, for the player of interest at $s$ to win. $m(s)$ is defined likewise, but for the opposite player. With no loss of generality, we focus on $n(s)$ for the remainder of this section, unless otherwise specified.

By definition, $n(s)$ is a perfect measures of resource efficiency for proving $s$. Unfortunately, in order to derive the *true* value of $n(s)$, we would need to traverse the game tree starting at $s$ by induction as follows. Suppose $s_T$ are terminal states. Then, $n(s_T) = 1$ if the player to play at $s$ is winning, and $n(s_T) = \infty$ otherwise. Suppose $s_{AND}$ are non-terminal AND nodes. Then, $n(s_{AND}) = 1 + \sum_{s_i} n(s_i)$, where $s_i$ are the children of $s_{AND}$. Suppose $s_{OR}$ are non-terminal OR nodes. Then, $n(s_{OR}) = 1 + \min_{s_i} n(s_i)$ for all children $s_i$. To obtain $\min_{s_i} n(s_i)$, we would need to search all children of non-terminal OR nodes; combined with having to search all children of non-terminal AND nodes, the conclusion is that the entire tree rooted at $s$ will have to be searched.

In practice, even if we cannot obtain the true value, a sufficiently accurate heuristic estimate of $n(s_i)$, which we denote as $\bar{n}(s_i)$, can be useful in guiding the search. In the ideal case, $\bar{n}(s_i)$ perfectly preserves the order of $n(s_i)$, i.e. $\bar{n}(s_1) \leq \bar{n}(s_2)$ if and only if $n(s_1) \leq n(s_2)$. Then, for $s_{OR}$, we would only have to compute the heuristic $\bar{n}(s_i)$ for all children $s_i$, then choose to search the child with the smallest value. The difficulty, as is always the case in heuristic search, is in designing such a heuristic so that $\bar{n}(s_i)$ preserves the order of $n(s_i)$ as best as possible.

While prioritization according to $\bar{n}$ is straightforward for OR nodes, the AND node case requires some more thought. Theoretically, in this case $\bar{n}$ should have no effect on priority since all children need to be searched for AND nodes. In practice, with an imperfect heuristic, the order in which children of AND nodes are searched impacts how quickly OR node prioritization mistakes are corrected.

To obtain reasonably accurate heuristics $\bar{n}$ and $\bar{m}$, we take advantage of AlphaZero's ability to converge on accurate positional value estimates through self-play, and modify the algorithm so that it approximates $n$ and $m$ instead. We describe the details on how the training targets are defined in subsection 3.2 and the training loop in subsection 3.3. The heuristics $\bar{n}$ and $\bar{m}$ are then used in a modified PNS algorithm, which we describe in subsection 3.4.

### 3.2 THE PROOF TREE SIZE ESTIMATE AS A TRAINING TARGET

Let us assume that $s$ is the root, is an OR node, has $b$ children, $s_1, ..., s_b$, and that all $\bar{n}(s_i)$ are available with $\bar{n}(s_1) \leq \bar{n}(s_2) \leq ... \leq \bar{n}(s_b)$. We propose that $\bar{n}(s) := \bar{n}(s_1)$, similar to the way PNs are calculated for OR nodes in PNS. Since $\bar{n}(s)$ grows exponentially, the $+1$ term's impact is overall negligible. Moreover, the omission of this term preserves the order of $\bar{n}$ among all children, so it can be safely omitted for simplicity. As described in subsection 3.1, the OR-player should prioritize searching $s_1$, or those $s_i$ with small $i$. The dual case where the root is an AND node is $\bar{n}(s) := \sum_{i=1}^{b} \bar{n}(s_i)$, where $s_i$ are the children of $s$, following PNs for AND nodes in PNS. In order to design the heuristic $\bar{n}(s)$ such that it preserves the ordering of $n(s)$ as accurately as possible, a naive way as describe in the previous subsection would be to perform brute force search to obtain all the ground truths for machine learning. However, it is apparent that this is not feasible given the exponential tree size growth.

Instead, we use a modified MCTS in the self-play phase of AlphaZero to collect game episodes, which are then used to learn $\bar{n}(s)$.

1. For the terminal state, the ground truths are set to 1 if the root player wins, and $\infty$ otherwise.
2. For all non-terminal OR nodes $s_t$ and its child $s_{t+1}$ in the episode, $\bar{n}(s_t)$ is set to $\bar{n}(s_{t+1})$.

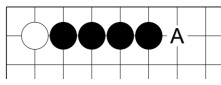

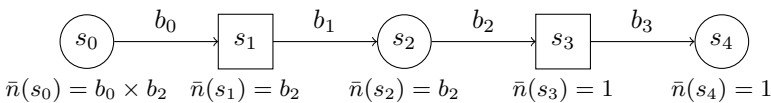

Figure 1: An illustration for different $b$.

Figure 2: Estimate $\bar{n}(s)$ from an episode of a game.

3. For all non-terminal AND nodes $s_t$ and its child $s_{t+1}$ in the episode, $\bar{n}(s_t)$ is set to $b \times \bar{n}(s_{t+1})$, where $b$ is the number of children for $s_t$.

We use $b \times \bar{n}(s_{t+1})$ for AND nodes as the training target since we do not know the values for all other siblings of $s_{t+1}$ (recall that self-play generates single episodes), which makes it difficult to obtain $\sum_{i=1}^{b} \bar{n}(s_i)$. Effectively, we are assuming that the costs for all of $s_t$'s children are equal to $s_{t+1}$'s. There are domain-specific improvements for many games to improve the accuracy of this estimate. As an example, the Gomoku position illustrated in Fig. 1 contains what is called a "four threat" (4T) for Black. We know that Black will achieve a 5-in-a-row and win if White does not block at A, so we can effectively reduce $b$ to 1.

To differentiate the case where we use heuristics to reduce $b$ like in the 4T example above, we will use $b_{heur}$ to specify the number of valid children when heuristics are used, and $b_{max}$ for the setting where we use the entire action space. We illustrate three examples for $b$ in 15x15 Gomoku. First, $b_{max}$ is always 225 since that is the maximum number of moves for the 15x15 board; next, if we use the game independent heuristic where we only consider legal moves, we set $b_{heur}$ to the number of empty grids left on the board; last, if a 4T exists, as in Fig. 1, $b_{heur} = 1$.

An example episode is illustrated in Fig. 2. It starts from $s_0$ and ends at $s_4$ with a win (achieving the goal) for the OR-player; $b_0$ to $b_3$ are the branching factors of $s_0$ to $s_3$ respectively (with no loss of generality to whether $b_{heur}$ or $b_{max}$ is used). The heuristic function $\bar{n}(s)$ for the states in the episode are set to the following: $\bar{n}(s_3) = \bar{n}(s_4) = 1$, $\bar{n}(s_1) = \bar{n}(s_2) = b_2 \times \bar{n}(s_3) = b_2$, and $\bar{n}(s_0) = b_0 \times \bar{n}(s_1) = b_0 \times b_2$. Based on this calculation, we can see that every move counting backwards from the terminal node increases the proof cost by a factor of $b$ equally when using $b_{max}$. Thus, the resulting heuristic can be seen as a measure of how many moves are left until the solution, so it can be useful for finding the shortest path to the solution.

There are some similarities and differences between our heuristic and the concept of PNs in PNS. First, PN/DN are dynamic quantities meant to change as PNS progresses and new nodes are expanded and evaluated. In contrast, $\bar{n}(s)$ is a static estimate of an oracle number that represents the minimum proof tree size. More specifically, rather than the PN value, $\bar{n}(s)$ estimates PN plus the current tree size of $s$.

Second, PNS makes no assumptions as to which player is more likely to win, and so both PN/DN are used. Since we already have self-play episodes to train from, we add an auxiliary target $\bar{m}(s)$, which is conceptually similar to DNs. In Fig. 2, $\bar{m}(s_0) = \bar{m}(s_1) = \bar{m}(s_2) = \bar{m}(s_3) = \bar{m}(s_4) = \infty$.

Last, in PNS, the search arrives at the MPN by choosing the minimum PN at OR nodes, and the minimum DN at AND nodes. In our method, while OR nodes behave similarly in that the smallest $\bar{n}$ value is chosen, for AND nodes, we prioritize moves which have the maximum $\bar{n}$ value. The OR node estimate is most accurate when the OR-player chooses the move with the lowest $\bar{n}(s)$ value because $n(s) = 1 + \min n(s_i)$. To leverage the AlphaZero algorithm, where the two players' goals run counter to each other, the AND-player behavior should be designed such that it tries its best to oppose the OR-player, which in this case means it should pick the largest $\bar{n}$. Intuitively, the OR-player will try to prove a win as quickly as possible, while the AND-player will delay as much as it can. It is important to stress that this difference only affects how the heuristic network is trained, not how search algorithms work. In other words, the choice of the maximum $\bar{n}$ value for AND nodes is used only in the self-play phase described in subsection 2.4 when training the heuristic network.

### 3.3 Training the Proof Cost Network with AlphaZero

We train the Proof Cost Network (PCN) $f_\theta(s) = (p, v_n, v_m)$, with parameters $\theta$, such that it predicts the policy distribution $p$, and the values $v_n$ and $v_m$, corresponding to the proof cost $\bar{n}(s)$ and auxil-

iary $\bar{m}(s)$ for the state $s$. Intuitively, for $p$, moves with higher probabilities lead to faster solutions for $s$, either proof or disproof. Our method replaces the AlphaZero $v(s)$ output, which estimate the win rate (or outcome of the game $z$), with $v_n(s)$ and $v_m(s)$, which estimate $\bar{n}(s)$ and the auxiliary target $\bar{m}(s)$. In practice, since $\bar{n}(s)$ grows exponentially the closer $s$ is to the root, we replace $\bar{n}(s)$ with the surrogate $l_n(s)$, where $l_n(s) = \log(\bar{n}(s))$, likewise with $\bar{m}(s)$.

With the single-sided proof cost heuristic $\bar{n}$, our method tends to perform stronger when we designate a specific player as the OR-player and attempt a proof (as opposed to solving problems without any assumptions as to the winner/loser). Thus, we need to first decide which are the AND/OR-players, e.g. knowing that Black is proven to be the winner for 15x15 Gomoku, we set Black as the OR-players. In the self-play phase, ideally, the OR-player should select $\arg\min_{a_i} \bar{n}(s_i)$ among all candidate moves $a_i$ which lead to $s_i$. Intuitively this motivates the self-play agent to solve $s$ as quickly as possible. Meanwhile, the AND-player should select $\arg\max_{a_i} \bar{n}(s_i)$ to prevent the OR-player from solving the game quickly, as described in subsection 3.2.

To implement this behavior into the MCTS process for move selection during self-play, we make the following changes. In the MCTS selection phase, we still follow eq. 1, but now $Q(s, a)$ is a moving average of $v_n(s)$, representing the estimated overall mean cost of solving $s$ with the action $a$. Since $Q(s, a)$ is not limited to $[-1, 1]$, we calculate $\bar{Q}(s, a)$ by normalizing $Q(s, a)$ with the following equation (similar to Schrittwieser et al. (2020)):

$$\bar{Q}(s, a) = 2 \times \frac{Q(s, a) - \min_{s,a \in Tree} Q(s, a)}{\max_{s,a \in Tree} Q(s, a) - \min_{s,a \in Tree} Q(s, a)} - 1. \tag{3}$$

Lastly, to motivate the OR/AND-player to choose small/large values of $\bar{n}$, we simply flip the value to $-\bar{Q}(s, a)$ for the OR-player.

During AlphaZero optimization, we randomly sample the self-play games and calculate $\bar{n}$ and $\bar{m}$ as described in subsection 3.2, then convert them to $l_n$ and $l_m$ accordingly. The network is optimized by a modified version of eq. 2, as follows:

$$L = \phi(v_n)^{\mathrm{T}} \log l_n + \phi(v_m)^{\mathrm{T}} \log l_m - \pi^{\mathrm{T}} \log p + c\|\theta\|^2, \tag{4}$$

where $\phi$ is a transformation function that changes scalar values to categorical representations (Schrittwieser et al., 2020).

### 3.4 FOCUSED DFPN WITH PROOF COST NETWORK

In this subsection, we present a method that incorporates PCN into the FDFPN solver (Henderson et al., 2009), called FDFPN-PCN. With the PCN, since the value outputs $v_n$ and $v_m$ are not scalars between $[-1, 1]$, we cannot directly apply them as heuristics like FDFPN-CNN (Gao et al., 2017), described in subsection 2.3.

We propose a separate method based on FDFPN and FDFPN-CNN that is able to use the PCN as a heuristic in the FDFPN solver. The fundamental search behaves just as DFPN would, including PN/DN calculation and the MPN selection mechanism, with two differences. First, we follow FDFPN-CNN where the policy output serves as move ordering, and FDFPN where the branching factor is limited by a parameter $\mu$. Second, since $v_n$ and $v_m$ take inspiration from PN/DN, we can initialize PN/DN values with the PCN as follows. Let us assume we are expanding state $s$, at which point we will call $f_\theta(s)$. A naive way would be to set $PN = v_n(s)$ and $DN = v_m(s)$ directly. The problem is that we do not have $v_n(s_i)$ and $v_m(s_i)$ for every expanded child. Additionally, calling $f_\theta(s_i)$ for all $s_i$ slows the entire search down. To conserve time, we initialize the PN/DN of $s_i$, then calculate the PN/DN of $s$ with its children's values. If $s$ is an OR node, we set $PN = v_n(s)$ and $DN = v_m(s)/b$ for all $s_i$, where $b$ is the number of children for $s$. On the other hand, if $s$ is an AND node, we set $PN = v_n(s)/b$ and $DN = v_m(s)$ for all $s_i$. This way, the PN/DN of $s$ will then be $v_n(s)$ and $v_m(s)$, respectively.

## 4 EXPERIMENTS

### 4.1 TRAINING SETTINGS

All experiments were performed by solving 15x15 Gomoku and 9x9 Killall-Go problems. We use two different *solvers*, one MCTS and the other FDFPN. The MCTS solver is an AlphaZero player

with AND/OR tree logic to propagate win/loss information. The FDFPN solver follows the mechanism described in 3.4. Both solvers make use of transposition tables (Nelson, 1985) to conserve evaluation costs when identical positions are encountered during search[3].

For each experiment, we focus on comparing the number of solved problems by using different *networks* as heuristics on the same solvers. There are four network configurations: (1) no network, (2) $\alpha 0$ (AlphaZero), (3) PCN-$b_{max}$, and (4) PCN-$b_{heur}$. For fairness, we use the same training settings for each network configuration, described as follows. We run 400 MCTS simulations for each move during self-play, for a total of 1,500,000 games, and the network is optimized every 5,000 games. The network contains 5 residual blocks with 64 filters, is optimized by SGD with 0.9 for momentum, 1e-4 for weight decay, and a fixed learning rate of 0.02. We use 1080Ti GPUs for training, where the network is implemented with PyTorch (Paszke et al., 2019). Each 15x15 Gomoku training takes about 1,000 GPU-Hours, and for 9x9 Killall-Go, about 1,500 GPU-Hours. For simplicity, we select the last snapshot model for each network in our experiments.

Additional detailed settings are as follows. In 15x15 Gomoku, the network input has 4 channels, including black and white stones in the current position, and two channels representing the turn color. The $\alpha$ for Dirichlet noise is set to 0.05. The maximum value ($\infty$), used to set training targets, is set to 1600. There are two *variations* for 15x15 Gomoku: (1) NDK: no domain knowledge is provided, and (2) 4T: uses the four threat heuristic. For the NDK variation, all legal moves are considered in the solver; $b_{max} = 225$, and $b_{heur}$ is the number of legal moves. For the 4T variation in Gomoku, whenever a 4T occurs, the solver will only consider blocking[4]; $b_{max} = 225$; $b_{heur} = 1$ with a 4T and is the number of legal moves else wise. Note that $b_{max}$ and $b_{heur}$ are not "branching factors" for the solver, but hyperparameters that define the training labels for $\bar{n}$ and $\bar{m}$.

9x9 Killall-Go (Cauwet et al., 2015) is similar to the game of Go except the following additional rules. Black plays four stones first on the board as shown in Fig. 4a, then White and Black play normally, as in standard Go. White wins if any white stones becomes unconditionally alive, otherwise Black wins. Namely, Black wins if and only if there are no white stones on the board at the end of the game. The network input for 9x9 Killall-Go has 18 channels, including black and white stones for the last 8 positions, and two channels representing the turn color. The $\alpha$ for Dirichlet noise is set to 0.2. The maximum value ($\infty$) is set to 500. In 9x9 Killall-Go, we implement two heuristics for the solver to help the game end quickly; first is Benson's algorithm (Benson, 1976) which detects unconditional life, which we use to end the game immediately if White achieves unconditional life anywhere on the board[5]; second, we prohibit players to fill their own true eyes. There are also two variations for Go, one assuming White is the OR-player, and the other Black; recall that $v_n$ is the proof cost estimate for the OR-player. For both variations, $b_{max}$ is set to 82 ($9 \times 9 + 1$ for passing), and $b_{heur}$ is set to the number of legal moves excluding filling in one's own eye.

### 4.2 Solving 15x15 Gomoku problems with MCTS-based Solver

A set of 77 problems are generated automatically for the experiment, detailed in Appendix B. The results for the number of solved problems within 30 minutes are shown in Table 1. We list the node counts in Appendix C. From the table, when no domain knowledge is provided for 15x15 Gomoku, both PCN-$b_{max}$ and PCN-$b_{heur}$ outperform $\alpha 0$, with PCN-$b_{max}$ being better than PCN-$b_{heur}$.

|  | No Network | $\alpha 0$ | PCN-$b_{max}$ | PCN-$b_{heur}$ |
|---|---|---|---|---|
| 15x15 Gomoku (NDK) | 1 / 77 | 23 / 77 | **43 / 77** | 38 / 77 |
| 15x15 Gomoku (4T) | 22 / 77 | 64 / 77 | **77 / 77** | 73 / 77 |
| 9x9 Killall-Go (OR: White) | 1 / 81 | 28 / 81 | **79 / 81** | 76 / 81 |
| 9x9 Killall-Go (OR: Black) | | | 38 / 81 | **46 / 81** |

Table 1: The number of problems that can be solved by MCTS-based solver within 30 minutes.

---

[3]Note that we only consider the single-ko as repetition in 9x9 Killall-Go, as was the case in van der Werf et al. (2003) on solving small Go boards.

[4]Since this changes the way the game behaves by removing all other branches, the networks training will also be affected.

[5]Like 4T, this affects the game tree, and therefore also the network training process.

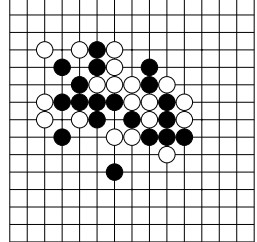
(a) A 15x15 Gomoku problem.

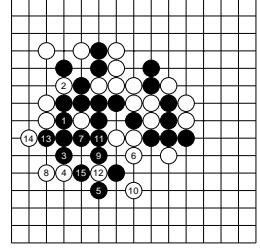
(b) Solving with PCN-$b_{heur}$.

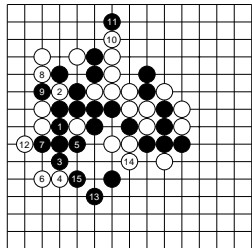
(c) Solving with PCN-$b_{max}$.

Figure 3: An illustration for using PCNs on solving 15x15 Gomoku problems.

We use Fig. 3, one of the problems, to illustrate that with the help of 4T, PCN-$b_{heur}$ can take advantage of the VCF (victory by continuous four) (gom, 2021) strategy more effectively. Fig. 3a is a problem where Black plays first and wins. Fig. 3b shows the solution following $b_{heur}$ calculation, where Black wins using VCF. Namely, if White does not respond to any of Black's threats, Black wins directly. In $b_{heur}$, $l(s) \approx 0$ since the branching factor is reduced to 1 by 4T. Fig. 3c shows the solution found by PCN-$b_{max}$, which has the same length as PCN-$b_{heur}$, but move 7 (Black) is not a 4T. This means that White has many choices, such as moves 8 and 10, that does not result in an immediate loss. All possibilities need to be searched to ensure a Black win, so limiting White choices narrows down the search and improves efficiency. With $b_{max}$, since the branching factor is always set to 225, the solver cannot identify 4Ts. As a result, PCN-$b_{max}$ requires 7,090 nodes while PCN-$b_{heur}$ only uses 2,283 to solve this problem. We expect that PCN-$b_{heur}$ can perform even better when incorporating more domain knowledge, such as three-threats.

## 4.3 Solving 9x9 Killall-Go problems with MCTS-based Solver

Next, we try to solve 81 automatically generated problems for 9x9 Killall-Go. We exclude Black wins from the generated problem set because they are trivial for PCN for the following reason. The goal for Black in this game is to own the entire board. However, since we generate problems from self-play outcomes, for Black winning states close to the end of the game, e.g. in Fig. 4b, Black has many similar moves that lead to victory. In situations like these, AlphaZero-based agents are known to play passively, unnecessarily extending the game length, which is a well-known problem for MCTS-based players. In fact, the AlphaZero-based Leela Chess Zero program implemented a "moves left head" in its network to handle this issue (Forstén & Pascutto, 2019), which functions similarly to $\bar{n}$ when we use $b_{max}$. Whereas the "moves left head" is only used near the end of the game, PCN is used throughout a problem. As favorable as this is for PCN, it is not a fair comparison, so we excluded Black win problems from the Killall-Go set.

Now we observe results for White win problems. Both $b_{max}$ and $b_{heur}$ still outperform $\alpha0$ as shown in Table 1. Note that since the baseline and $\alpha0$ do not need to designate which side is the OR-player, they only have one result each. Interestingly, for the variation where Black is set to be the OR-player (and $v_m$ is used instead of $v_n$ as the proof cost), it can still solve about half of the problems, despite the problem being White win and $v_m$ being an auxiliary task.

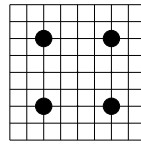
(a) 9x9 Killall-Go opening.

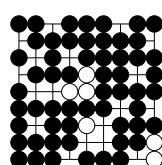
(b) A black win problem.

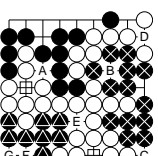
(c) A white win problem.

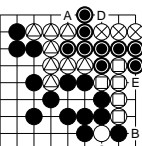
(d) A white win problem.

Figure 4: An illustration for using PCN and AlphaZero on solving 9x9 Killall-Go problems.

Fig. 4 shows two examples to illustrate the difference between $\alpha0$ and our method. In Fig. 4c, a problem for White to play and win, White can become Benson safe by directly playing at A, which

forms two eyes (square labels). Both PCN-$b_{max}$ and PCN-$b_{heur}$ only focus on A, which solves the problem directly. However, since White has many ways to win in this position, $\alpha0$ will attempt other moves, e.g. play at B, C, or D to kill the cross-marked black stones, or play at E, F, or G to kill the triangle-mark stones. Since the goal for AlphaZero is to win, and all of A to G can reach this goal, $\alpha0$ does not focus on A only, spending more nodes to solve this problem.

In the problem in Fig. 4d, there is a capturing race between the three white blocks (marked as cross, triangle, and square) and the black block (marked as circle). White can win the capturing race by directly playing at A (for Black D, reply E, and for E, reply D). For another sure win, White can first extend liberties for the square-marked white blocks by making an *exchange*, a common technique in Go, where White plays at B and Black replies at C. For both winning moves at A and B, we observe that $\alpha0$ tends to make the exchange at B first to increase the win rate, while PCN prefers the faster A. Again, exploring all possibilities is costly. So much so that $\alpha0$ takes nearly three times the node count of PCN to solve this problem.

## 4.4 SOLVING WITH THE FDFPN SOLVER

Table 2 shows the results for baseline FDFPN[6] and the three types of networks. In FDFPN, we set $\mu = 0.1$ on both 15x15 Gomoku and 9x9 Killall-Go. The result shows that both PCN-$b_{max}$ and PCN-$b_{heur}$ still outperform $\alpha0$. When comparing MCTS and FDFPN solvers directly, we find that there is no clear winner. FDFPN can solve the same number of problems for 9x9 Killall-Go, while MCTS-based solver is slightly better for 15x15 Gomoku. The fact that neither DFPN nor MCTS dominates the other in solving has been corroborated by previous research (Ewalds, 2012; Wei et al., 2015). However, in either case, PCN has a positive impact on efficiency. We therefore expect PCN to be extendable to other search algorithms.

|  | No Network | $\alpha0$ | PCN-$b_{max}$ | PCN-$b_{heur}$ |
|---|---|---|---|---|
| 15x15 Gomoku (NDK) | 6 / 77 | 15 / 77 | 45 / 77 | **48 / 77** |
| 15x15 Gomoku (4T) | 34 / 77 | 41 / 77 | **71 / 77** | 69 / 77 |
| 9x9 Killall-Go (OR: White) | 31 / 81 | 76 / 81 | **79 / 81** | 77 / 81 |
| 9x9 Killall-Go (OR: Black) |  |  | 66 / 81 | 68 / 81 |

Table 2: The number of problems that can be solved by FDFPN solver within 30 minutes.

## 5 DISCUSSION

The experiments demonstrate that, in general, AlphaZero-like networks can be used to enhance solving without expert knowledge. It is somewhat surprising that $b_{heur}$ is not consistently better than $b_{max}$, since it is theoretically a more accurate estimate. A possible explanation is that the main benefit of PCN is contingent upon the ordering of $\bar{n}$, rather than $\bar{n}$ itself, and that the limited scope of heuristics may have led to inaccuracies in general. However, there is potential in fine-tuning a combination of heuristics so that $b_{heur}$ performs even better. We leave this as a future direction of research. We note that 15x15 Gomoku has been solved by a combination of PNS, dependency-based search, and threat-space search (Allis, 1994). Nonetheless, we choose to perform experiments on Gomoku because it is a straight-forward benchmark that is well-known and often used in games. By excluding the highly specific dependency-based and threat-space search in our experiments, and by including experiments for the yet unsolved 9x9 Killall-Go, we wish to demonstrate PCN's generality and positive impact on solving efficiency, regardless of whether expert knowledge is provided. Combining PCN with powerful, highly game-specific heuristics is left as a future topic of research. A strong case that supports the idea that neural networks do not clash with hand-crafted heuristics in solving is the results presented by Gao et al. (2017), where a highly game-specific engine was combined with networks to enhance solving on 8x8 Hex. We expect that PCN can be used to solve more challenging game problems in the future with the introduction of more sophisticated knowledge. Lastly, PCN may be helpful for domains in which heuristics can be difficult to design, e.g. planning, chemistry, material science, among others.

---

[6]FDFPN relies on networks to provide heuristics for progressive widening and move orderings, so without a network, the baseline is implemented as DFPN.

**Ethics Statement**    We do not foresee any potential for ethical concerns for this research.

**Reproducibility Statement**    All experiments can be reproduced by following the instructions in the README file on https://github.com/kds285/proof-cost-network, including source code, problem sets, MCTS and FDFPN solvers, and the trained models used in this paper.

## ACKNOWLEDGEMENT

This research is partially supported by the Ministry of Science and Technology (MOST) of Taiwan under Grant Numbers 110-2634-F-009-022, 110-2634-F-A49-004 and 110-2221-E-A49-067-MY3, and the computing resources are partially supported by National Center for High-performance Computing (NCHC) of Taiwan. The authors would also like to thank Professor Martin Müller and anonymous reviewers for their valuable comments.

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

## A  Playing Strength

Although our method does not optimize for strongest playing strength, it is an interesting question worth investigating, since we expect playing and solving to be correlated tasks. With PCN the OR-player tends to solve (win) the game as soon as possible, which coincides with strong play. On the other hand, the goal for the AND-player is to maximum $\bar{n}(s)$; it receives the maximum reward ($\infty$) if it wins the game. This encourages the AND-player to win the game if possible, and prevent the OR-player from winning the game as much as possible otherwise. Overall, PCN optimization preserves playing strength while prioritizing game solving.

Interestingly, the results in Table 3 show that our method is stronger than $\alpha 0$ for 9x9 Killall-Go (with a win rate of 60%), and has nearly the same playing strength as $\alpha 0$ in 15x15 Gomoku. Each experiment contains 250 games with alternating Black and White assignment.

|  |  | Win Rate (WR) | Black WR | White WR |
|---|---|---|---|---|
| 15x15 Gomoku (NDK) | PCN-$b_{max}$ | 50.40% $\pm$ 6.21% | 100.00% | 0.80% |
| 15x15 Gomoku (NDK) | PCN-$b_{heur}$ | 50.80% $\pm$ 6.21% | 100.00% | 1.60% |
| 15x15 Gomoku (4T) | PCN-$b_{max}$ | 50.40% $\pm$ 6.21% | 96.80% | 4.00% |
| 15x15 Gomoku (4T) | PCN-$b_{heur}$ | 50.00% $\pm$ 6.21% | 100.00% | 0.00% |
| 9x9 Killall-Go | PCN-$b_{max}$ (OR: White) | 50.00% $\pm$ 6.21% | 34.40% | 65.60% |
| 9x9 Killall-Go | PCN-$b_{heur}$ (OR: White) | 60.80% $\pm$ 6.07% | 66.40% | 55.20% |
| 9x9 Killall-Go | PCN-$b_{max}$ (OR: Black) | 62.25% $\pm$ 6.03% | 60.00% | 64.52% |
| 9x9 Killall-Go | PCN-$b_{heur}$ (OR: Black) | 60.08% $\pm$ 6.06% | 53.23% | 66.94% |

Table 3: Comparing the playing strength between PCNs and $\alpha 0$.

## B  Problem Set Generation

We generate self-play games to create the problem sets. For each self-play game, we start from the last move at the end of game and try to solve the game by using the three networks ($\alpha 0$, PCN-$b_{max}$, PCN-$b_{heur}$) with an MCTS solver. If the problem can be solved within 5 minutes, we rollback two moves and try solving the resulting larger problem. To generate problems with strong discriminating power, we repeat this process until one of the three programs cannot solve the problem within 5 minutes, upon which the problem is added into the problem set.

For 15x15 Gomoku, we choose Yixin (Sun, 2018), a strong Gomoku program that won several championships from 2012 to 2018, as the problem generating program. Self-play games were generated with one second per move. Since 15x15 Gomoku has been proven as Black win (Allis, 1994), and Yixin plays almost optimally, we only select problems with Black wins. We collect 77 problems for Black wins from two variations, 43 and 34 problems from the NDK and 4T respectively. Generally, the 34 problems is harder than 43 problems.

For 9x9 Killall-Go, since there are no open-source 9x9 Killall-Go programs, we simply use $\alpha 0$ and PCN-$b_{max}$ to generate self-play games. The self-play games were generated with 400 simulations per move. In addition, the Dirichlet noise is maintained in order to increase the diversity of self-play games. We collect 81 problems for White wins, 39 and 42 problems from the self-play games by $\alpha 0$ and PCN-$b_{max}$ respectively. Black wins are trivial for PCN but difficult for $\alpha 0$ (as we mention in subsection 4.3), so we exclude them from our problem set.

## C  Details for Solving Problem Sets with Different Solvers

Table 4 to 7 and Table 8 to 11 list the node count and the time for solving 15x15 Gomoku and 9x9 Killall-Go with different network settings as heuristics on both MCTS-based and FDFPN solvers. In the table, the fewest node counts and the shortest time to solve the problems among the different settings are bolded, where dashes represent settings where the problems cannot be solved within 30 minutes. Note that the "no network" configuration searches more nodes than any of the configurations where networks are used, within the same amount of time. Each solver runs with one CPU and one NVIDIA Tesla V100. In 15x15 Gomoku, the solver without networks can visit nearly

10,000 and 30,000 nodes on MCTS-based and FDFPN solvers respectively, while the solver with networks can only visit 600 and 1,000 nodes on MCTS-based and FDFPN solvers respectively. In 9x9 Killall-Go, the solver without networks can visit nearly 5,600 and 15,000 nodes on MCTS-based and FDFPN solvers respectively, while the solver with networks can only visit 600 and 950 nodes on MCTS-based and FDFPN solvers respectively. Despite this massive difference in node counts, network-assisted solvers can still solve more problems.

To further improve network-assisted solvers, we can use asynchronous evaluation between CPU and GPU. By doing so, the solver no longer needs to wait for the GPU to compute during evaluation. The FDFPN-CNN Hex solver by Gao et al. (2017) is one such example. In regards to the slower CNN evaluations, they state that the runtime overhead of CNN evaluation is not an issue, since the sophisticated hand-crafted Hex heuristics can be up to 10 times slower than the CNN evaluations in early positions. As a result, FDFPN-CNN can solve problems more quickly than DFPN and FDFPN (both without networks) in the same amount of time.

For 15x15 Gomoku, problems 1 to 43 are generated by NDK and problems 44 to 77 are generated by 4T. The NDK problem set is determined to be easier to solve than the 4T set because all NDK problems are solvable by the three networks trained with 4T, except for problem 6.

| ID | NDK | | | | 4T | | | |
|---|---|---|---|---|---|---|---|---|
| | No Network | $\alpha0$ | PCN-$b_{max}$ | PCN-$b_{heur}$ | No Network | $\alpha0$ | PCN-$b_{max}$ | PCN-$b_{heur}$ |
| 1 | - | - | 903,544 | **836,419** | - | 4,492 | **3,051** | 3,164 |
| 2 | - | 1,012,028 | **686,546** | 749,374 | 8,480,868 | 2,437 | 2,938 | **2,416** |
| 3 | - | - | **783,765** | 940,395 | 9,422,154 | 3,014 | **1,962** | 2,015 |
| 4 | - | 908,688 | **635,155** | 652,825 | 512,488 | **2,571** | 2,668 | 2,706 |
| 5 | - | - | **687,892** | 758,434 | - | 5,501 | **4,063** | 5,815 |
| 6 | - | 653,312 | **638,560** | - | - | 699,398 | **10,735** | - |
| 7 | - | 1,119,700 | 470,906 | **433,757** | 2,781,342 | 2,219 | **1,288** | 1,981 |
| 8 | - | **534,685** | 591,610 | 957,519 | 5,518,358 | 2,382 | 3,503 | **2,006** |
| 9 | - | - | **515,801** | 751,947 | 12,757,192 | **2,700** | 2,707 | 3,608 |
| 10 | - | **862,561** | 938,374 | - | 1,201,265 | 5,936 | 4,232 | **2,851** |
| 11 | - | 559,413 | **361,723** | - | - | **1,681** | 4,215 | 2,460 |
| 12 | - | **94,659** | 839,179 | - | 2,266,157 | 8,307 | 7,798 | **3,484** |
| 13 | - | 1,113,649 | **287,016** | 386,683 | - | 2,206 | 1,796 | **1,268** |
| 14 | - | - | **685,446** | 715,306 | - | **2,868** | 5,105 | 3,452 |
| 15 | - | - | 672,121 | **644,224** | 2,662,073 | 3,777 | **1,680** | 2,571 |
| 16 | - | 680,569 | **568,632** | 698,349 | 5,666,572 | **4,090** | 6,236 | 4,254 |
| 17 | - | 926,956 | **784,247** | 809,173 | - | 3,112 | 2,949 | **2,878** |
| 18 | - | 1,032,936 | **639,561** | 1,014,679 | - | 8,837 | **2,234** | 13,235 |
| 19 | - | 1,000,189 | **93,057** | 273,953 | - | 18,424 | 7,090 | **2,283** |
| 20 | - | 990,392 | **781,816** | 935,157 | 507,186 | **2,782** | 2,799 | 2,802 |
| 21 | - | - | **651,847** | 830,393 | - | 5,679 | **4,735** | 6,405 |
| 22 | - | 1,030,986 | 770,577 | **639,290** | 264,069 | **2,577** | 2,776 | 3,043 |
| 23 | - | - | 450,445 | **431,117** | - | 4,631 | 1,320 | **1,281** |
| 24 | - | - | **420,275** | 454,366 | - | 5,271 | **1,758** | 3,367 |
| 25 | - | **395,492** | 987,041 | - | 6,959,837 | **1,618** | 2,738 | 8,177 |
| 26 | - | 1,156,791 | **653,604** | 757,952 | - | 4,576 | **1,717** | 3,054 |
| 27 | - | - | 712,891 | **509,584** | 11,033,791 | 5,159 | 4,675 | **2,758** |
| 28 | - | 1,103,090 | 862,540 | **796,669** | 13,049,802 | **3,423** | 4,066 | 3,786 |
| 29 | - | - | 861,855 | **834,637** | 293,187 | **2,789** | 3,867 | 3,538 |
| 30 | - | - | **652,067** | 865,387 | 198,562 | 4,097 | **2,097** | 2,464 |
| 31 | - | 231,797 | **69,152** | 1,028,435 | 1,808,145 | 5,014 | **3,957** | 4,035 |
| 32 | - | - | **172,368** | 577,034 | - | **20,016** | 78,047 | 92,014 |
| 33 | - | - | **530,040** | 648,157 | - | 6,658 | **1,775** | 1,944 |
| 34 | - | - | **487,206** | 504,550 | - | 3,368 | **1,254** | 1,257 |
| 35 | - | - | **399,652** | 593,615 | - | 6,204 | **1,788** | 2,849 |
| 36 | - | - | **603,235** | 666,346 | - | 8,290 | **2,770** | 6,285 |
| 37 | 13,955,507 | **465,375** | 799,030 | 953,092 | 93,510 | 1,273 | **682** | 983 |
| 38 | - | - | **583,851** | 646,149 | 13,734,784 | 3,771 | 3,198 | **2,951** |
| 39 | - | 1,106,087 | 747,578 | **719,157** | 2,091,312 | 2,348 | 2,938 | **2,115** |
| 40 | - | 1,036,870 | **466,665** | 807,696 | - | 1,865 | 1,897 | **1,861** |
| 41 | - | - | **539,523** | 548,735 | 11,774,261 | 3,964 | 1,384 | **983** |
| 42 | - | 1,049,955 | **719,127** | 912,464 | - | 158,302 | 2,622 | **2,345** |
| 43 | - | - | 889,495 | **762,337** | - | 5,004 | 3,134 | **2,751** |

| ID | NDK | | | | 4T | | | |
|----|-----------|----|----------------|-----------------|------------|-----------|----------------|-----------------|
|    | No Network | α0 | PCN-$b_{max}$ | PCN-$b_{heur}$ | No Network | α0 | PCN-$b_{max}$ | PCN-$b_{heur}$ |
| 44 | - | - | - | - | - | 948,612 | **11,816** | 136,827 |
| 45 | - | - | - | - | - | - | 32,691 | **28,290** |
| 46 | - | - | - | - | - | - | **63,209** | 364,122 |
| 47 | - | - | - | - | - | **233,314** | 294,051 | 549,238 |
| 48 | - | - | - | - | - | 660,045 | 657,619 | **651,482** |
| 49 | - | - | - | - | - | 879,833 | **536,679** | 584,436 |
| 50 | - | - | - | - | - | - | 418,675 | **209,113** |
| 51 | - | - | - | - | - | 1,008,722 | **35,214** | 61,311 |
| 52 | - | - | - | - | - | - | **478,926** | 911,914 |
| 53 | - | - | - | - | - | - | **61,331** | 317,466 |
| 54 | - | - | - | - | - | - | 273,710 | **249,321** |
| 55 | - | - | - | - | - | 633,981 | 479,537 | **453,190** |
| 56 | - | - | - | - | - | - | 268,991 | **238,730** |
| 57 | - | - | - | - | - | 854,630 | 287,420 | **284,574** |
| 58 | - | - | - | - | - | 1,147,125 | **464,559** | 891,960 |
| 59 | - | - | - | - | - | **261,682** | 581,801 | 361,861 |
| 60 | - | - | - | - | - | 553,491 | **72,787** | 587,483 |
| 61 | - | - | - | - | - | 535,287 | **328,709** | 1,005,042 |
| 62 | - | - | - | - | - | 802,585 | **585,090** | - |
| 63 | - | - | - | - | - | - | 831,631 | **294,646** |
| 64 | - | - | - | - | - | 1,042,594 | **342,084** | 411,711 |
| 65 | - | - | - | - | - | 199,122 | **120,191** | - |
| 66 | - | - | - | - | - | - | 342,094 | **304,327** |
| 67 | - | - | - | - | - | **84,050** | 794,435 | 150,296 |
| 68 | - | - | - | - | - | **198,359** | 438,366 | 866,356 |
| 69 | - | - | - | - | - | - | **376,675** | 489,782 |
| 70 | - | - | - | - | - | 527,975 | **505,992** | 719,737 |
| 71 | - | - | - | - | - | 310,169 | **267,580** | 654,385 |
| 72 | - | - | - | - | - | - | 538,784 | **272,023** |
| 73 | - | - | - | - | - | - | 354,198 | **304,270** |
| 74 | - | - | - | - | - | 1,048,393 | **415,890** | 437,995 |
| 75 | - | - | - | - | - | 95,530 | **77,530** | - |
| 76 | - | - | - | - | - | **329,171** | 928,163 | 675,018 |
| 77 | - | - | - | - | - | - | **544,247** | 673,300 |

Table 4: The number of nodes for solving 15x15 Gomoku problems by MCTS-based solver within 30 minutes.

| ID | No Network | $\alpha 0$ | OR: White | | OR: Black | |
|---|---|---|---|---|---|---|
| | | | PCN-$b_{max}$ | PCN-$b_{heur}$ | PCN-$b_{max}$ | PCN-$b_{heur}$ |
| 1 | - | - | 437,427 | **339,633** | - | 785,525 |
| 2 | - | - | **4,618** | 33,160 | - | - |
| 3 | - | - | **192,309** | 211,574 | - | - |
| 4 | - | - | 6,555 | **5,379** | 10,642 | 12,223 |
| 5 | - | 1,133,362 | 554,846 | **233,965** | 877,935 | 1,061,766 |
| 6 | - | - | 156,314 | **70,436** | - | - |
| 7 | - | - | **349,935** | 421,157 | - | 925,416 |
| 8 | - | - | **409,362** | 449,030 | - | - |
| 9 | - | - | **160,034** | 201,496 | - | - |
| 10 | - | - | 437,794 | **429,521** | - | - |
| 11 | - | - | 529,023 | **400,903** | - | - |
| 12 | - | - | **156,672** | 182,602 | 586,437 | 440,626 |
| 13 | - | - | 403,117 | 405,541 | **397,002** | 454,119 |
| 14 | - | - | 69,294 | **50,143** | - | - |
| 15 | - | 61,204 | **46,534** | 355,033 | 55,525 | - |
| 16 | - | - | 455,218 | **405,530** | - | 497,717 |
| 17 | - | - | **227,845** | 274,924 | - | 791,601 |
| 18 | - | 666,273 | **576,607** | 605,017 | - | 693,361 |
| 19 | - | - | **570,149** | 574,660 | - | 917,036 |
| 20 | - | - | 70,939 | **68,671** | 120,887 | 75,346 |
| 21 | - | - | 491,865 | **363,363** | - | - |
| 22 | - | - | 453,038 | **245,053** | - | 830,380 |
| 23 | - | - | 476,680 | **453,657** | - | - |
| 24 | - | **259,634** | 544,990 | - | - | - |
| 25 | - | - | 257,715 | **193,855** | 1,020,210 | - |
| 26 | 3,902,940 | - | **1,981** | 2,851 | 137,263 | 3,990 |
| 27 | - | - | 102,884 | 51,444 | **20,350** | 45,310 |
| 28 | - | 550,715 | 368,897 | **352,836** | 855,787 | 719,720 |
| 29 | - | - | **14,519** | 19,203 | 18,116 | 108,653 |
| 30 | - | - | 325,283 | **233,978** | 484,467 | 612,807 |
| 31 | - | - | 340,825 | **321,689** | - | 536,506 |
| 32 | - | 1,072,229 | 650,540 | 506,066 | **483,832** | 507,081 |
| 33 | - | 699,976 | 670,141 | 658,158 | 790,459 | **647,781** |
| 34 | - | 1,152,164 | 136,569 | **94,909** | 182,727 | 237,083 |
| 35 | - | 682,934 | **210,127** | 330,116 | 549,703 | 335,215 |
| 36 | - | 889,594 | 362,456 | **327,290** | 606,410 | 470,420 |
| 37 | - | 690,280 | 319,203 | **214,199** | 241,502 | 249,145 |
| 38 | - | 660,857 | 782,389 | 656,469 | **604,494** | 836,375 |
| 39 | - | **188,863** | 1,014,805 | 214,094 | 458,387 | 285,908 |

| ID | No Network | $\alpha0$ | OR: White | | OR: Black | |
|---|---|---|---|---|---|---|
| | | | PCN-$b_{max}$ | PCN-$b_{heur}$ | PCN-$b_{max}$ | PCN-$b_{heur}$ |
| 40 | - | - | **391,510** | 646,069 | 590,645 | 723,964 |
| 41 | - | - | **381,749** | 425,425 | - | 757,229 |
| 42 | - | - | **603,583** | 722,377 | - | - |
| 43 | - | - | **360,083** | 385,016 | - | - |
| 44 | - | 835,478 | **134,934** | 372,879 | - | - |
| 45 | - | - | **49,328** | 94,117 | - | - |
| 46 | - | - | - | **1,016,020** | - | - |
| 47 | - | - | 766,115 | **618,161** | - | 833,902 |
| 48 | - | - | 456,166 | **289,890** | 980,939 | 700,542 |
| 49 | - | 598,013 | 313,586 | 321,995 | - | **205,000** |
| 50 | - | - | 661,909 | **499,206** | - | - |
| 51 | - | - | 544,187 | **455,409** | - | - |
| 52 | - | 1,058,102 | 458,325 | **237,500** | - | - |
| 53 | - | 1,079,923 | **71,166** | 127,477 | - | 625,432 |
| 54 | - | - | **165,955** | 210,692 | - | - |
| 55 | - | 524,162 | **258,051** | 270,910 | 418,062 | 406,587 |
| 56 | - | 681,781 | **294,556** | - | 416,460 | - |
| 57 | - | - | **429,972** | 763,296 | - | - |
| 58 | - | - | 361,453 | **278,488** | 664,460 | 646,122 |
| 59 | - | 702,626 | 176,605 | **123,341** | 600,530 | 323,570 |
| 60 | - | - | 263,338 | 279,139 | 431,096 | **249,955** |
| 61 | - | 656,951 | **10,355** | 187,801 | 207,440 | 121,996 |
| 62 | - | - | 306,116 | **241,333** | 1,032,826 | 958,818 |
| 63 | - | - | **545,717** | 782,683 | - | 779,757 |
| 64 | - | - | **353,096** | 438,834 | - | 421,204 |
| 65 | - | 952,956 | 478,213 | **463,806** | 615,836 | 566,792 |
| 66 | - | 979,685 | 748,465 | **654,854** | 709,501 | 804,058 |
| 67 | - | - | 815,722 | **462,405** | - | - |
| 68 | - | - | 896,922 | **547,063** | - | - |
| 69 | - | 773,437 | **85,371** | 97,659 | - | - |
| 70 | - | - | **536,289** | 656,174 | - | - |
| 71 | - | - | **758,350** | 800,725 | 913,483 | - |
| 72 | - | - | **305,553** | 352,893 | 544,303 | - |
| 73 | - | 761,161 | **403,713** | 445,774 | 680,177 | 553,268 |
| 74 | - | - | **564,561** | - | - | - |
| 75 | - | 830,266 | 37,573 | **34,223** | 594,617 | 161,846 |
| 76 | - | - | **315,803** | 332,915 | - | - |
| 77 | - | - | **541,111** | - | - | - |
| 78 | - | 712,686 | 553,105 | **451,014** | 892,304 | 606,382 |
| 79 | - | - | 779,636 | **320,105** | - | - |
| 80 | - | 333,202 | - | - | **144,310** | - |
| 81 | - | - | **305,263** | 480,568 | 528,225 | 543,474 |

Table 5: The number of nodes for solving 9x9 Killall-Go problems by MCTS-based solver within 30 minutes.

| ID | NDK | | | | 4T | | | |
|---|---|---|---|---|---|---|---|---|
| | No Network | $\alpha0$ | PCN-$b_{max}$ | PCN-$b_{heur}$ | No Network | $\alpha0$ | PCN-$b_{max}$ | PCN-$b_{heur}$ |
| 1 | - | - | 543,160 | **300,826** | 2,177,195 | 28,632 | 12,195 | **7,225** |
| 2 | - | 1,052,180 | **339,216** | 575,363 | 390,282 | 3,202 | **2,406** | 8,729 |
| 3 | - | - | **179,216** | 264,336 | - | - | **2,394** | 2,532 |
| 4 | - | - | 262,151 | **224,069** | 2,688,024 | 54,325 | 2,610 | **2,383** |
| 5 | - | - | 426,690 | **261,045** | 1,245,999 | 13,423 | **3,748** | 4,916 |
| 6 | - | - | 566,537 | **510,176** | - | - | 10,133 | **9,760** |
| 7 | - | 1,081,268 | 180,677 | **131,859** | 1,459,111 | 3,310 | **1,248** | 2,092 |
| 8 | - | 512,308 | 361,114 | **196,655** | 1,409,132 | 2,689 | 3,627 | **855** |
| 9 | - | - | 375,162 | **174,019** | - | 292,830 | **2,078** | 2,498 |
| 10 | - | - | **158,011** | 160,819 | 1,437,840 | 26,542 | 2,797 | **889** |
| 11 | - | **242,705** | 432,620 | 398,452 | 87,376 | **1,271** | 3,745 | 3,694 |
| 12 | 1,187,021 | **5,326** | 250,782 | 205,715 | 2,497 | **58** | 3,990 | 5,932 |
| 13 | - | - | 265,696 | **189,882** | 6,610,347 | 164,078 | **1,215** | 2,867 |
| 14 | - | - | **382,906** | 529,466 | 2,022,077 | 11,191 | **3,480** | 4,308 |
| 15 | - | - | **248,953** | 375,955 | 23,691,221 | 2,972 | **2,024** | 2,035 |
| 16 | - | - | 646,271 | **516,882** | 21,022,574 | 7,978 | **4,748** | 9,440 |
| 17 | - | - | **441,738** | 527,535 | - | 199,248 | 5,792 | **3,355** |
| 18 | - | - | 254,980 | **83,524** | - | 4,173 | **877** | 8,192 |
| 19 | 2,076,725 | **136,831** | 339,247 | 285,687 | 22,819 | **1,298** | 4,688 | 8,163 |
| 20 | - | - | **286,949** | 318,495 | 894,652 | 13,950 | **2,782** | 2,782 |
| 21 | - | - | 839,876 | **281,356** | - | - | **6,815** | 16,147 |
| 22 | - | - | 323,425 | **114,274** | 23,531,128 | 3,492 | **2,330** | 2,472 |
| 23 | 6,674,096 | **140,310** | 419,757 | 196,482 | 18,671 | **292** | 1,164 | 2,336 |
| 24 | - | - | 322,051 | **159,776** | 34,292,960 | 4,162 | **1,588** | 1,588 |
| 25 | - | - | 1,452,491 | **453,056** | - | - | **4,104** | 4,389 |
| 26 | - | - | 641,847 | **209,837** | 3,227,357 | **1,556** | 1,606 | 3,296 |
| 27 | - | 615,044 | 496,379 | **115,182** | 975,205 | 5,759 | **1,165** | 1,167 |
| 28 | - | - | 508,185 | **261,287** | - | - | **2,523** | 2,873 |
| 29 | - | 363,656 | 741,413 | **277,864** | 99,310 | **2,563** | 5,855 | 4,823 |
| 30 | - | - | 1,241,236 | **234,492** | 4,750,995 | 13,191 | **2,058** | 2,058 |
| 31 | 423,196 | 443,681 | 526,924 | **329,662** | 3,767 | **14** | 6,793 | 7,824 |
| 32 | 2,698,810 | 156,409 | **5,573** | 16,313 | 9,891 | **986** | 1,942 | 2,499 |
| 33 | - | - | **130,813** | 226,744 | - | 839,813 | **1,239** | 2,520 |
| 34 | - | - | 253,464 | **121,042** | 24,304,241 | 44,919 | **1,194** | 1,194 |
| 35 | - | 430,639 | **294,400** | 482,790 | 5,816,694 | **3,297** | 6,354 | 4,420 |
| 36 | - | - | 1,459,236 | **1,308,142** | 7,288,513 | 1,295,829 | 10,432 | **5,081** |
| 37 | 49,658,836 | 159,883 | 44,904 | **41,163** | 1,007,093 | 465 | **400** | 401 |
| 38 | - | - | **138,609** | 258,479 | - | - | **4,723** | 10,196 |
| 39 | - | 462,126 | 450,905 | **177,037** | 235,960 | 2,377 | **1,674** | 2,471 |
| 40 | - | - | **210,674** | 212,195 | 2,526,582 | 61,503 | 2,100 | **1,953** |
| 41 | - | 426,269 | 654,285 | **85,745** | 34,663 | 1,101 | **814** | 1,274 |
| 42 | - | - | 1,378,481 | **186,671** | 37,091,680 | 625,646 | **1,759** | 2,770 |
| 43 | - | - | 645,077 | **206,283** | - | 4,295 | 3,642 | **2,042** |

| ID | NDK | | | | 4T | | | |
|---|---|---|---|---|---|---|---|---|
| | No Network | $\alpha 0$ | PCN-$b_{max}$ | PCN-$b_{heur}$ | No Network | $\alpha 0$ | PCN-$b_{max}$ | PCN-$b_{heur}$ |
| 44 | - | - | - | **1,792,382** | - | - | **251,433** | 267,422 |
| 45 | - | - | - | - | - | - | 56,306 | **44,743** |
| 46 | - | - | 541,211 | **183,799** | 34,104,016 | 439,562 | 37,161 | **4,239** |
| 47 | - | - | - | - | - | - | **93,820** | 159,739 |
| 48 | - | - | - | - | - | - | **81,989** | 1,790,298 |
| 49 | - | - | - | - | - | - | 480,072 | **386,461** |
| 50 | - | - | - | - | - | - | - | **1,610,384** |
| 51 | - | - | - | - | - | - | **298,338** | 1,033,424 |
| 52 | - | - | - | - | - | - | - | **1,798,644** |
| 53 | - | - | - | - | - | - | **88,042** | 125,743 |
| 54 | - | - | - | - | - | - | - | - |
| 55 | - | - | - | - | - | - | 356,504 | **337,247** |
| 56 | - | - | - | - | - | - | - | - |
| 57 | - | - | - | - | - | - | - | - |
| 58 | - | - | - | - | - | - | **948,526** | 1,584,855 |
| 59 | - | - | - | - | - | 1,438,345 | **468,174** | 982,546 |
| 60 | - | - | - | - | - | - | **92,478** | 114,131 |
| 61 | - | - | - | - | - | - | **1,068,376** | - |
| 62 | - | - | - | - | - | - | **519,382** | 958,805 |
| 63 | - | - | - | - | - | - | 594,017 | **281,991** |
| 64 | - | - | - | - | - | - | **460,714** | 597,067 |
| 65 | - | - | - | - | - | - | 1,081,090 | **578,086** |
| 66 | - | - | - | - | 21,638,030 | - | **1,300,876** | - |
| 67 | - | - | - | **1,374,985** | - | 69,420 | 57,288 | **26,993** |
| 68 | - | - | - | - | - | - | **611,155** | 1,490,409 |
| 69 | - | - | - | - | - | - | **709,441** | 1,297,859 |
| 70 | - | - | **654,994** | - | - | - | **60,078** | 1,507,901 |
| 71 | - | - | - | - | - | - | 705,697 | **318,132** |
| 72 | - | - | - | **1,822,097** | - | - | **102,770** | 251,269 |
| 73 | - | - | - | - | - | - | - | - |
| 74 | - | - | - | - | - | - | **709,832** | - |
| 75 | - | - | - | - | - | - | **143,401** | 204,409 |
| 76 | - | - | - | - | - | - | **1,359,561** | - |
| 77 | - | - | - | **1,257,882** | - | 130,946 | 88,314 | **34,050** |

Table 6: The number of nodes for solving 15x15 Gomoku problems by FDFPN solver within 30 minutes.

| ID | No Network | $\alpha 0$ | OR: White | | OR: Black | |
|---|---|---|---|---|---|---|
| | | | PCN-$b_{max}$ | PCN-$b_{heur}$ | PCN-$b_{max}$ | PCN-$b_{heur}$ |
| 1 | - | **105,581** | - | 251,446 | - | - |
| 2 | 440,072 | 1,185,075 | **85,304** | 849,589 | - | 1,517,593 |
| 3 | - | 165,180 | 66,414 | **48,006** | 282,534 | 254,917 |
| 4 | 156,629 | **11,862** | 25,744 | 58,257 | 402,391 | 598,018 |
| 5 | 12,225,417 | 134,212 | **49,372** | 82,856 | 899,547 | 384,364 |
| 6 | **94,776** | - | 229,484 | - | - | - |
| 7 | - | 88,568 | 49,381 | **42,336** | 327,931 | 92,959 |
| 8 | - | 324,258 | 99,992 | **84,942** | 550,406 | 1,204,383 |
| 9 | - | 91,815 | **38,332** | 58,377 | 904,472 | 576,223 |
| 10 | - | - | **168,602** | - | - | - |
| 11 | - | **76,156** | 707,040 | 439,179 | - | - |
| 12 | - | 109,511 | **47,928** | 65,455 | 166,016 | 158,398 |
| 13 | - | 91,202 | **68,399** | 70,416 | 78,761 | 69,916 |
| 14 | 913,926 | 139,042 | **30,826** | 38,837 | 344,542 | - |
| 15 | 714,249 | - | 594,300 | - | **440,596** | - |
| 16 | - | 74,575 | **56,884** | 73,200 | 716,529 | 649,376 |
| 17 | - | 104,483 | 67,225 | **52,809** | 269,910 | 326,266 |
| 18 | - | **371,360** | 404,028 | 764,740 | - | 994,024 |
| 19 | - | 236,576 | **153,104** | 175,386 | 538,322 | 454,310 |
| 20 | - | **49,697** | 52,844 | 81,449 | 840,154 | 52,794 |
| 21 | - | 229,227 | 414,353 | **112,112** | - | - |
| 22 | - | 68,046 | 352,552 | **37,635** | 349,411 | 85,562 |
| 23 | - | 201,110 | 154,734 | **108,834** | - | 854,835 |
| 24 | - | **55,827** | - | 92,276 | 1,152,654 | 735,299 |
| 25 | 5,751,181 | **70,464** | 72,376 | 73,656 | 228,755 | 151,235 |
| 26 | **18,955** | 155,636 | 308,769 | 119,415 | 311,918 | 210,555 |
| 27 | 267,915 | **17,387** | 30,025 | 24,954 | 26,811 | 27,820 |
| 28 | - | 94,007 | 77,754 | **73,580** | 1,073,040 | 246,813 |
| 29 | 160,073 | 20,682 | 4,142 | **4,140** | 25,690 | 32,209 |
| 30 | 6,306,634 | **19,652** | 41,903 | 41,139 | 116,303 | 74,733 |
| 31 | - | 123,465 | 176,841 | **61,750** | 184,350 | 575,231 |
| 32 | - | 204,057 | 77,380 | **74,878** | 432,883 | 207,227 |
| 33 | - | 453,994 | 455,168 | **142,337** | 368,747 | 292,738 |
| 34 | 9,370,462 | 76,096 | 169,268 | **40,835** | 173,576 | - |
| 35 | 18,455,057 | 92,739 | 125,019 | **90,691** | 959,719 | 207,097 |
| 36 | - | 130,105 | 187,389 | **115,271** | - | 555,793 |
| 37 | - | 365,501 | 191,485 | **39,745** | 41,332 | 59,091 |
| 38 | - | 171,239 | **75,797** | 92,581 | 153,552 | 96,992 |
| 39 | 1,325,967 | **40,338** | 309,778 | 243,186 | 451,765 | 61,551 |

| ID | No Network | $\alpha 0$ | OR: White | | OR: Black | |
|---|---|---|---|---|---|---|
| | | | PCN-$b_{max}$ | PCN-$b_{heur}$ | PCN-$b_{max}$ | PCN-$b_{heur}$ |
| 40 | 1,568,427 | **11,062** | 68,513 | 62,227 | 121,919 | 119,334 |
| 41 | 5,582,584 | 85,341 | 77,923 | **49,357** | 239,248 | 80,325 |
| 42 | 12,159,258 | **208,083** | 250,075 | 727,195 | 1,515,988 | 297,450 |
| 43 | - | 398,535 | **104,223** | 109,207 | 155,762 | 393,325 |
| 44 | 7,868,261 | **51,794** | 116,171 | 95,303 | 105,250 | 127,852 |
| 45 | - | 42,763 | **26,045** | 59,236 | 260,910 | 1,155,942 |
| 46 | - | **332,714** | 481,330 | 470,446 | 517,647 | 481,537 |
| 47 | - | 153,737 | 178,395 | 324,031 | **105,871** | 649,895 |
| 48 | 3,546,708 | **61,985** | 190,729 | 63,435 | 570,575 | 119,572 |
| 49 | 3,538,421 | - | 60,819 | **33,260** | 542,081 | 363,988 |
| 50 | - | 119,373 | 59,956 | **57,896** | 172,871 | 226,773 |
| 51 | - | 202,341 | 86,683 | **75,376** | - | 1,194,333 |
| 52 | - | 60,529 | **27,602** | 623,511 | 932,927 | - |
| 53 | 2,532,381 | 931,158 | 49,522 | **43,069** | - | 141,410 |
| 54 | 7,796,804 | 96,960 | **28,603** | 38,995 | 127,005 | 216,658 |
| 55 | 23,427,876 | 78,237 | 44,023 | **42,123** | 51,499 | 61,907 |
| 56 | - | 62,164 | **58,695** | 229,139 | 84,872 | 230,724 |
| 57 | - | **297,229** | 298,112 | 563,293 | 1,277,324 | 567,529 |
| 58 | - | **45,637** | 67,159 | 60,974 | 131,614 | 174,553 |
| 59 | - | - | **35,303** | - | - | 247,154 |
| 60 | - | **76,577** | 96,680 | 134,284 | 396,421 | 99,898 |
| 61 | 3,228,036 | 180,397 | 1,000,430 | 551,266 | **45,367** | 1,004,425 |
| 62 | 1,732,495 | 95,734 | 67,390 | **53,681** | 84,854 | 67,360 |
| 63 | - | 136,273 | **51,063** | 86,901 | 164,990 | 98,916 |
| 64 | 19,265,809 | **94,870** | 199,416 | 193,159 | 212,520 | 166,488 |
| 65 | - | 226,771 | 81,651 | 83,622 | **81,599** | 308,324 |
| 66 | - | 101,958 | **52,661** | 169,573 | 53,265 | 64,568 |
| 67 | - | 191,855 | 120,125 | **92,385** | 948,000 | - |
| 68 | - | 453,862 | 130,535 | 247,408 | **128,594** | 143,000 |
| 69 | 3,538,272 | **40,611** | 389,813 | 268,809 | 1,212,621 | 761,402 |
| 70 | - | 184,262 | **141,848** | 226,816 | - | 754,497 |
| 71 | - | 345,270 | 338,292 | **135,537** | 323,659 | 952,482 |
| 72 | - | 113,105 | **63,087** | 491,455 | - | - |
| 73 | - | 241,979 | **108,124** | 112,510 | 231,873 | 1,488,998 |
| 74 | - | 329,613 | 478,603 | **108,690** | 338,535 | 707,935 |
| 75 | 4,266,281 | 39,189 | 177,188 | **26,459** | 61,456 | 108,976 |
| 76 | - | 216,528 | **162,290** | 222,421 | 516,084 | 591,543 |
| 77 | 5,212,057 | 74,484 | 372,793 | 283,878 | **57,896** | - |
| 78 | - | 101,740 | 890,386 | 1,313,460 | **75,293** | 214,671 |
| 79 | 3,951,997 | 913,206 | **538,557** | 1,560,344 | - | - |
| 80 | 5,556,840 | **51,665** | 189,705 | 137,227 | 688,728 | 431,534 |
| 81 | - | 203,609 | 258,430 | **116,320** | 250,402 | 130,868 |

Table 7: The number of nodes for solving 9x9 Killall-Go problems by FDFPN solver within 30 minutes.

| ID | NDK | | | | 4T | | | |
|---|---|---|---|---|---|---|---|---|
| | No Network | $\alpha 0$ | PCN-$b_{max}$ | PCN-$b_{heur}$ | No Network | $\alpha 0$ | PCN-$b_{max}$ | PCN-$b_{heur}$ |
| 1 | - | - | 1,641.85 | **1,391.13** | - | 6.91 | **5.24** | 5.26 |
| 2 | - | 1,603.53 | **1,164.22** | 1,211.55 | 909.63 | **3.79** | 4.76 | 4.06 |
| 3 | - | - | **1,285.74** | 1,575.19 | 960.39 | 4.81 | **3.42** | 3.56 |
| 4 | - | 1,359.59 | **1,086.16** | 1,089.31 | 49.73 | **4.16** | 4.56 | 4.62 |
| 5 | - | - | **1,156.20** | 1,300.91 | - | 8.29 | **6.80** | 9.62 |
| 6 | - | **985.13** | 1,111.49 | - | - | 1,043.69 | **17.64** | - |
| 7 | - | 1,679.10 | 762.32 | **713.26** | 303.18 | 3.62 | **2.41** | 3.44 |
| 8 | - | **818.23** | 996.76 | 1,617.30 | 635.09 | 3.79 | 5.92 | **3.52** |
| 9 | - | - | 872.64 | 1,236.99 | 1,508.31 | **4.40** | 4.60 | 6.11 |
| 10 | - | **1,302.95** | 1,588.18 | - | 117.27 | 8.89 | 6.91 | **5.00** |
| 11 | - | 849.36 | **612.68** | - | - | **2.74** | 7.08 | 4.18 |
| 12 | - | **143.59** | 1,459.04 | - | 252.64 | 12.48 | 12.49 | **5.89** |
| 13 | - | 1,764.95 | **473.21** | 643.69 | - | 4.20 | 3.13 | **2.37** |
| 14 | - | - | **1,390.14** | 1,407.12 | - | **4.36** | 8.31 | 5.74 |
| 15 | - | - | 1,145.41 | **1,110.29** | 276.29 | 5.82 | **2.92** | 4.35 |
| 16 | - | 1,046.80 | **970.73** | 1,155.58 | 619.56 | **6.34** | 10.14 | 7.06 |
| 17 | - | 1,361.17 | **1,341.06** | 1,425.94 | - | **4.90** | 4.95 | 4.94 |
| 18 | - | 1,556.42 | **1,133.75** | 1,747.26 | - | 13.35 | **3.89** | 21.70 |
| 19 | - | 1,494.58 | **156.03** | 457.72 | - | 26.66 | 11.88 | **3.89** |
| 20 | - | 1,471.13 | **1,323.64** | 1,587.48 | 49.84 | **4.34** | 4.76 | 4.77 |
| 21 | - | - | **1,132.25** | 1,422.36 | - | 8.45 | **7.92** | 10.42 |
| 22 | - | 1,549.52 | 1,296.35 | **1,055.47** | 25.47 | **4.38** | 4.65 | 5.08 |
| 23 | - | - | 753.05 | **727.39** | - | 7.14 | **2.41** | 2.42 |
| 24 | - | - | **711.89** | 755.30 | - | 8.29 | **3.07** | 5.81 |
| 25 | - | **606.42** | 1,654.57 | - | 685.59 | **2.62** | 4.63 | 13.23 |
| 26 | - | 1,739.03 | **1,103.72** | 1,237.65 | - | 7.01 | **3.04** | 5.15 |
| 27 | - | - | 1,193.00 | **827.29** | 1,191.56 | 7.78 | 7.68 | **4.65** |
| 28 | - | 1,701.56 | 1,480.14 | **1,336.36** | 1,550.17 | **5.14** | 6.75 | 7.60 |
| 29 | - | - | 1,478.13 | **1,373.58** | 24.06 | **4.41** | 6.50 | 5.83 |
| 30 | - | - | **1,051.20** | 1,428.38 | 18.52 | 6.42 | **3.76** | 4.20 |
| 31 | - | 353.11 | **115.50** | 1,722.87 | 161.98 | 7.75 | **6.60** | 6.67 |
| 32 | - | - | **292.02** | 1,004.17 | - | **29.57** | 127.56 | 144.17 |
| 33 | - | - | 887.65 | 1,196.47 | - | 9.90 | **3.10** | 3.38 |
| 34 | - | - | 789.29 | 825.19 | - | 5.21 | **2.31** | 2.35 |
| 35 | - | - | **680.09** | 988.54 | - | 9.47 | **3.20** | 5.00 |
| 36 | - | - | **1,011.33** | 1,127.02 | - | 14.79 | **4.76** | 10.34 |
| 37 | 1,546.59 | **727.96** | 1,367.75 | 1,539.05 | 7.41 | 2.17 | **1.41** | 1.89 |
| 38 | - | - | **944.65** | 1,051.17 | 1,444.89 | 5.76 | 5.35 | **5.04** |
| 39 | - | 1,655.04 | 1,255.56 | **1,201.15** | 210.16 | 3.81 | 5.01 | **3.68** |
| 40 | - | 1,553.98 | **798.03** | 1,346.00 | - | **3.02** | 3.40 | 3.32 |
| 41 | - | - | 910.09 | **900.26** | 1,339.80 | 7.40 | 2.51 | **1.95** |
| 42 | - | 1,588.72 | **1,444.92** | 1,498.58 | - | 282.98 | 5.35 | **4.05** |
| 43 | - | - | 1,654.17 | **1,261.58** | - | 7.69 | 5.39 | **4.72** |

| ID | NDK | | | | 4T | | | |
|---|---|---|---|---|---|---|---|---|
| | No Network | $\alpha0$ | PCN-$b_{max}$ | PCN-$b_{heur}$ | No Network | $\alpha0$ | PCN-$b_{max}$ | PCN-$b_{heur}$ |
| 44 | - | - | - | - | - | 1,416.51 | **18.58** | 219.25 |
| 45 | - | - | - | - | - | - | 53.10 | **47.68** |
| 46 | - | - | - | - | - | - | **107.05** | 609.18 |
| 47 | - | - | - | - | - | **342.41** | 483.62 | 934.34 |
| 48 | - | - | - | - | - | **963.28** | 1,295.06 | 1,106.29 |
| 49 | - | - | - | - | - | 1,287.97 | **908.03** | 1,001.88 |
| 50 | - | - | - | - | - | - | 730.09 | **354.13** |
| 51 | - | - | - | - | - | 1,477.58 | **56.31** | 103.56 |
| 52 | - | - | - | - | - | - | **848.60** | 1,605.55 |
| 53 | - | - | - | - | - | - | **103.88** | 556.51 |
| 54 | - | - | - | - | - | - | 437.53 | **404.02** |
| 55 | - | - | - | - | - | 915.69 | 791.14 | **742.92** |
| 56 | - | - | - | - | - | - | 436.69 | **390.40** |
| 57 | - | - | - | - | - | 1,292.87 | **478.50** | 491.27 |
| 58 | - | - | - | - | - | 1,765.34 | **820.44** | 1,565.14 |
| 59 | - | - | - | - | - | **382.00** | 1,114.89 | 623.94 |
| 60 | - | - | - | - | - | 823.72 | **117.10** | 1,002.04 |
| 61 | - | - | - | - | - | 836.36 | **566.13** | 1,788.09 |
| 62 | - | - | - | - | - | 1,181.13 | **998.12** | - |
| 63 | - | - | - | - | - | - | 1,475.81 | **510.46** |
| 64 | - | - | - | - | - | 1,569.05 | **554.86** | 691.11 |
| 65 | - | - | - | - | - | 307.41 | **204.81** | - |
| 66 | - | - | - | - | - | - | 573.36 | **503.69** |
| 67 | - | - | - | - | - | **127.16** | 1,592.37 | 258.61 |
| 68 | - | - | - | - | - | **296.08** | 772.00 | 1,534.01 |
| 69 | - | - | - | - | - | - | **646.82** | 835.70 |
| 70 | - | - | - | - | - | **770.25** | 872.30 | 1,241.65 |
| 71 | - | - | - | - | - | 456.92 | **447.29** | 1,099.47 |
| 72 | - | - | - | - | - | - | 948.82 | **477.99** |
| 73 | - | - | - | - | - | - | 574.38 | **495.11** |
| 74 | - | - | - | - | - | 1,564.80 | **680.49** | 739.09 |
| 75 | - | - | - | - | - | 145.03 | **129.98** | - |
| 76 | - | - | - | - | - | **488.09** | 1,541.00 | 1,129.12 |
| 77 | - | - | - | - | - | - | **931.98** | 1,120.48 |

Table 8: The number of seconds for solving 15x15 Gomoku problems by MCTS-based solver within 30 minutes.

| ID | No Network | $\alpha 0$ | OR: White | | OR: Black | |
|---|---|---|---|---|---|---|
| | | | PCN-$b_{max}$ | PCN-$b_{heur}$ | PCN-$b_{max}$ | PCN-$b_{heur}$ |
| 1 | - | - | 717.29 | **550.25** | - | 1,308.06 |
| 2 | - | - | **7.64** | 53.68 | - | - |
| 3 | - | - | **310.45** | 455.50 | - | - |
| 4 | - | - | 10.95 | **8.91** | 17.15 | 19.88 |
| 5 | - | 1,699.97 | 883.03 | **373.53** | 1,407.40 | 1,716.12 |
| 6 | - | - | 257.29 | **112.90** | - | - |
| 7 | - | - | **593.42** | 692.15 | - | 1,574.53 |
| 8 | - | - | **678.70** | 760.72 | - | - |
| 9 | - | - | **249.15** | 318.21 | - | - |
| 10 | - | - | **720.91** | 743.88 | - | - |
| 11 | - | - | 859.67 | **670.63** | - | - |
| 12 | - | - | **251.54** | 314.89 | 960.04 | 701.50 |
| 13 | - | - | 657.45 | 637.39 | **624.48** | 728.20 |
| 14 | - | - | 112.69 | **80.58** | - | - |
| 15 | - | 92.62 | **76.34** | 589.04 | 90.74 | - |
| 16 | - | - | 865.33 | **670.93** | - | 803.47 |
| 17 | - | - | **361.44** | 454.74 | - | 1,283.96 |
| 18 | - | 1,016.10 | **947.34** | 980.16 | - | 1,141.83 |
| 19 | - | - | 930.83 | **927.06** | - | 1,562.77 |
| 20 | - | - | 114.96 | **110.13** | 194.30 | 135.16 |
| 21 | - | - | 859.63 | **594.28** | - | - |
| 22 | - | - | 710.73 | **408.30** | - | 1,404.40 |
| 23 | - | - | 776.83 | **745.13** | - | - |
| 24 | - | **397.35** | 923.48 | - | - | - |
| 25 | - | - | 409.48 | **306.63** | 1,665.77 | - |
| 26 | 694.75 | - | **3.43** | 4.87 | 219.22 | 6.67 |
| 27 | - | - | 166.86 | 88.92 | **32.27** | 73.49 |
| 28 | - | 813.48 | 590.50 | **576.55** | 1,420.97 | 1,189.86 |
| 29 | - | - | **23.39** | 30.57 | 29.27 | 171.38 |
| 30 | - | - | 516.90 | **376.76** | 766.32 | 976.99 |
| 31 | - | - | 562.35 | **525.96** | - | 885.91 |
| 32 | - | 1,642.09 | 1,051.68 | 848.82 | **777.79** | 948.32 |
| 33 | - | 1,055.87 | 1,089.96 | **1,047.25** | 1,287.27 | 1,057.17 |
| 34 | - | 1,786.24 | 226.78 | **151.43** | 293.71 | 380.98 |
| 35 | - | 1,047.99 | **337.09** | 533.51 | 876.26 | 545.96 |
| 36 | - | 1,413.90 | 699.19 | **527.28** | 983.93 | 749.81 |
| 37 | - | 1,026.93 | 525.22 | **349.81** | 382.59 | 393.80 |
| 38 | - | 1,021.81 | 1,319.52 | 1,045.71 | **993.30** | 1,335.96 |
| 39 | - | **286.99** | 1,639.58 | 346.67 | 753.18 | 486.64 |

| ID | No Network | $\alpha 0$ | OR: White | | OR: Black | |
|---|---|---|---|---|---|---|
| | | | PCN-$b_{max}$ | PCN-$b_{heur}$ | PCN-$b_{max}$ | PCN-$b_{heur}$ |
| 40 | - | - | **609.87** | 1,055.59 | 972.74 | 1,154.97 |
| 41 | - | - | **624.47** | 789.08 | - | 1,225.60 |
| 42 | - | - | **969.97** | 1,151.57 | - | - |
| 43 | - | - | **581.76** | 726.57 | - | - |
| 44 | - | 1,254.88 | **214.95** | 600.15 | - | - |
| 45 | - | - | **82.95** | 151.06 | - | - |
| 46 | - | - | - | **1,691.70** | - | - |
| 47 | - | - | 1,244.11 | **989.85** | - | 1,345.12 |
| 48 | - | - | 747.32 | **478.10** | 1,620.92 | 1,119.28 |
| 49 | - | 965.99 | 499.94 | 523.49 | - | **336.27** |
| 50 | - | - | 1,083.70 | **819.99** | - | - |
| 51 | - | - | 902.41 | **734.51** | - | - |
| 52 | - | 1,606.60 | 742.37 | **375.71** | - | - |
| 53 | - | 1,709.32 | **113.59** | 208.31 | - | 1,059.81 |
| 54 | - | - | **270.97** | 334.19 | - | - |
| 55 | - | 769.36 | **426.23** | 432.22 | 682.90 | 650.82 |
| 56 | - | 1,043.89 | **473.24** | - | 683.99 | - |
| 57 | - | - | **703.08** | 1,228.99 | - | - |
| 58 | - | - | 593.03 | **459.56** | 1,099.31 | 1,071.77 |
| 59 | - | 1,095.48 | 306.37 | **195.31** | 950.30 | 529.00 |
| 60 | - | - | 422.89 | 449.17 | 699.77 | **404.72** |
| 61 | - | 1,000.77 | **17.10** | 314.45 | 354.66 | 195.99 |
| 62 | - | - | 495.79 | **377.44** | 1,659.71 | 1,544.58 |
| 63 | - | - | **853.85** | 1,253.60 | - | 1,316.05 |
| 64 | - | - | **582.64** | 727.92 | - | 674.43 |
| 65 | - | 1,435.88 | 786.31 | **730.44** | 984.65 | 943.31 |
| 66 | - | 1,444.08 | 1,227.15 | **1,057.24** | 1,163.36 | 1,333.39 |
| 67 | - | - | 1,339.81 | **739.78** | - | - |
| 68 | - | - | 1,503.63 | **961.99** | - | - |
| 69 | - | 1,184.15 | **140.69** | 162.64 | - | - |
| 70 | - | - | **856.74** | 1,087.59 | - | - |
| 71 | - | - | **1,205.52** | 1,314.98 | 1,496.59 | - |
| 72 | - | - | **501.21** | 578.36 | 882.80 | - |
| 73 | - | 1,219.56 | **634.80** | 736.14 | 1,077.52 | 899.74 |
| 74 | - | - | **903.23** | - | - | - |
| 75 | - | 1,269.77 | 60.95 | **55.17** | 954.76 | 267.81 |
| 76 | - | - | **512.92** | 540.63 | - | - |
| 77 | - | - | **989.07** | - | - | - |
| 78 | - | 1,096.09 | 873.19 | **720.92** | 1,461.39 | 996.62 |
| 79 | - | - | 1,322.26 | **556.49** | - | - |
| 80 | - | 496.93 | - | - | **237.71** | - |
| 81 | - | - | **503.55** | 796.72 | 873.12 | 887.82 |

Table 9: The number of seconds for solving 9x9 Killall-Go problems by MCTS-based solver within 30 minutes.

| ID | NDK | | | | 4T | | | |
|---|---|---|---|---|---|---|---|---|
| | No Network | $\alpha 0$ | PCN-$b_{max}$ | PCN-$b_{heur}$ | No Network | $\alpha 0$ | PCN-$b_{max}$ | PCN-$b_{heur}$ |
| 1 | - | - | 452.58 | **250.79** | 92.40 | 34.36 | 14.90 | **10.23** |
| 2 | - | 1,089.85 | **287.76** | 486.31 | 19.31 | **5.97** | 6.27 | 13.83 |
| 3 | - | - | **147.90** | 219.14 | - | - | 5.95 | **4.52** |
| 4 | - | - | 225.64 | **187.48** | 112.61 | 70.03 | 4.27 | **4.24** |
| 5 | - | - | 373.72 | **214.83** | 57.89 | 17.62 | **6.50** | 6.62 |
| 6 | - | - | 483.94 | **426.65** | - | - | **13.08** | 13.44 |
| 7 | - | 1,050.54 | 150.95 | **111.42** | 67.48 | 5.39 | **3.35** | 3.98 |
| 8 | - | 500.92 | 290.77 | **174.87** | 64.05 | 4.71 | 5.57 | **3.01** |
| 9 | - | - | 315.60 | **148.64** | - | 353.89 | **3.96** | 4.40 |
| 10 | - | - | 131.21 | **129.81** | 62.44 | 29.35 | 4.42 | **3.10** |
| 11 | - | **228.54** | 381.68 | 344.47 | 6.07 | **3.10** | 5.48 | 5.29 |
| 12 | 38.44 | **8.55** | 204.32 | 171.83 | 2.44 | **2.28** | 7.15 | 7.14 |
| 13 | - | - | 225.16 | **158.58** | 277.23 | 161.44 | **3.27** | 5.08 |
| 14 | - | - | **386.61** | 428.26 | 87.17 | 16.28 | **4.99** | 5.81 |
| 15 | - | - | **204.07** | 343.96 | 995.42 | 6.11 | **3.90** | 3.93 |
| 16 | - | - | 553.34 | **422.95** | 935.03 | 12.68 | **6.53** | 9.81 |
| 17 | - | - | **362.46** | 538.79 | - | 265.95 | 7.08 | **5.06** |
| 18 | - | - | 213.73 | **69.44** | - | 7.56 | **3.39** | 13.55 |
| 19 | **59.33** | 169.88 | 289.70 | 244.17 | **2.69** | 4.76 | 6.78 | 9.13 |
| 20 | - | - | **234.84** | 279.29 | 41.60 | 16.44 | **4.69** | 5.61 |
| 21 | - | - | 701.95 | **230.92** | - | - | **10.40** | 18.68 |
| 22 | - | - | 267.92 | **94.72** | 952.18 | 6.51 | **4.23** | 4.37 |
| 23 | 196.64 | 171.50 | 339.06 | **160.70** | 2.71 | **2.58** | 3.18 | 4.25 |
| 24 | - | - | 267.21 | **130.39** | 1,500.09 | 6.97 | **3.50** | 3.56 |
| 25 | - | - | 1,344.21 | **373.23** | - | - | 6.21 | **5.90** |
| 26 | - | - | 533.24 | **175.49** | 137.86 | 4.40 | **3.55** | 5.78 |
| 27 | - | 572.92 | 400.28 | **96.45** | 43.25 | 8.08 | **3.15** | 3.15 |
| 28 | - | - | 419.59 | **217.19** | - | - | **4.55** | 4.88 |
| 29 | - | 309.18 | 645.21 | **236.40** | 6.10 | **4.11** | 7.44 | 6.59 |
| 30 | - | - | 1,082.56 | **193.67** | 218.66 | 16.67 | 4.00 | **3.86** |
| 31 | **13.94** | 389.29 | 447.95 | 272.70 | 2.45 | **2.18** | 8.07 | 9.72 |
| 32 | 77.56 | 150.14 | **7.54** | 17.33 | **2.26** | 3.83 | 4.25 | 5.70 |
| 33 | - | - | **108.39** | 187.57 | - | 1,036.16 | **3.24** | 4.62 |
| 34 | - | - | 229.58 | **102.62** | 1,047.46 | 48.76 | 3.40 | **3.31** |
| 35 | - | 500.48 | **244.87** | 396.00 | 257.96 | 6.10 | 9.04 | **5.85** |
| 36 | - | - | 1,476.63 | **1,135.60** | 259.40 | 1,704.00 | 13.61 | **6.96** |
| 37 | 1,708.60 | 159.03 | 39.99 | **35.99** | 45.19 | 2.53 | 2.72 | **2.51** |
| 38 | - | - | **115.46** | 257.76 | - | - | **8.38** | 12.98 |
| 39 | - | 462.48 | 462.54 | **168.51** | 12.81 | 4.15 | **3.53** | 4.46 |
| 40 | - | - | 178.03 | **173.06** | 105.34 | 68.63 | **4.25** | 5.05 |
| 41 | - | 415.11 | 543.92 | **72.02** | 3.55 | 3.50 | 4.04 | **3.23** |
| 42 | - | - | 1,322.47 | **152.56** | 1,748.07 | 712.34 | **4.53** | 6.47 |
| 43 | - | - | 534.58 | **169.33** | - | 7.06 | 7.06 | **4.01** |

| ID | NDK | | | | 4T | | | |
|---|---|---|---|---|---|---|---|---|
| | No Network | $\alpha 0$ | PCN-$b_{max}$ | PCN-$b_{heur}$ | No Network | $\alpha 0$ | PCN-$b_{max}$ | PCN-$b_{heur}$ |
| 44 | - | - | - | **1,531.60** | - | - | **231.69** | 254.88 |
| 45 | - | - | - | - | - | - | 54.78 | **44.81** |
| 46 | - | - | 479.45 | **154.43** | 1,365.49 | 552.57 | 35.40 | **6.32** |
| 47 | - | - | - | - | - | - | **93.92** | 147.53 |
| 48 | - | - | - | - | - | - | **73.80** | 1,535.73 |
| 49 | - | - | - | - | - | - | 428.07 | **355.17** |
| 50 | - | - | - | - | - | - | - | **1,467.75** |
| 51 | - | - | - | - | - | - | **260.30** | 975.85 |
| 52 | - | - | - | - | - | - | - | **1,573.53** |
| 53 | - | - | - | - | - | - | **82.38** | 122.28 |
| 54 | - | - | - | - | - | - | - | - |
| 55 | - | - | - | - | - | - | 323.44 | **303.80** |
| 56 | - | - | - | - | - | - | - | - |
| 57 | - | - | - | - | - | - | - | - |
| 58 | - | - | - | - | - | - | **834.94** | 1,385.13 |
| 59 | - | - | - | - | - | 1,747.38 | **422.92** | 1,119.64 |
| 60 | - | - | - | - | - | - | **81.03** | 98.32 |
| 61 | - | - | - | - | - | - | **932.23** | - |
| 62 | - | - | - | - | - | - | **452.09** | 857.88 |
| 63 | - | - | - | - | - | - | 550.55 | **274.11** |
| 64 | - | - | - | - | - | - | **426.04** | 559.13 |
| 65 | - | - | - | - | - | - | 1,260.36 | **558.35** |
| 66 | - | - | - | - | 929.66 | - | 1,194.21 | - |
| 67 | - | - | - | **1,146.29** | - | 81.16 | 50.39 | **27.80** |
| 68 | - | - | - | - | - | - | **583.33** | 1,385.67 |
| 69 | - | - | - | - | - | - | **614.95** | 1,148.70 |
| 70 | - | - | 559.80 | - | - | - | **52.98** | 1,421.51 |
| 71 | - | - | - | - | - | - | 606.93 | **280.18** |
| 72 | - | - | - | **1,531.69** | - | - | **98.30** | 229.78 |
| 73 | - | - | - | - | - | - | - | - |
| 74 | - | - | - | - | - | - | **609.68** | - |
| 75 | - | - | - | - | - | - | **144.79** | 215.34 |
| 76 | - | - | - | - | - | - | **1,159.34** | - |
| 77 | - | - | - | **1,075.57** | - | 149.06 | 87.45 | **34.11** |

Table 10: The number of seconds for solving 15x15 Gomoku problems by FDFPN solver within 30 minutes.

| ID | No Network | $\alpha0$ | OR: White | | OR: Black | |
|---|---|---|---|---|---|---|
| | | | PCN-$b_{max}$ | PCN-$b_{heur}$ | PCN-$b_{max}$ | PCN-$b_{heur}$ |
| 1 | - | **123.57** | - | 240.92 | - | - |
| 2 | **31.58** | 1,427.51 | 83.20 | 942.16 | - | 1,695.97 |
| 3 | - | 198.97 | 62.60 | **46.13** | 291.05 | 253.62 |
| 4 | **13.43** | 15.10 | 25.76 | 56.57 | 417.85 | 601.96 |
| 5 | 886.54 | 156.85 | **49.55** | 80.30 | 1,144.13 | 406.89 |
| 6 | **7.89** | - | 250.80 | - | - | - |
| 7 | - | 103.60 | 49.37 | **41.54** | 318.74 | 89.22 |
| 8 | - | 371.78 | 98.65 | **81.89** | 581.53 | 1,339.79 |
| 9 | - | 109.20 | **37.78** | 55.22 | 900.34 | 587.29 |
| 10 | - | - | **153.40** | - | - | - |
| 11 | - | **84.11** | 725.33 | 463.33 | - | - |
| 12 | - | 130.78 | **46.56** | 61.71 | 156.09 | 152.95 |
| 13 | - | 114.65 | **65.08** | 71.93 | 77.01 | 68.35 |
| 14 | 61.45 | 144.57 | **32.69** | 37.78 | 334.21 | - |
| 15 | **54.44** | - | 624.53 | - | 431.13 | - |
| 16 | - | 85.17 | **56.61** | 73.32 | 732.51 | 680.38 |
| 17 | - | 119.74 | 65.52 | **51.85** | 261.32 | 314.09 |
| 18 | - | 434.44 | **385.28** | 766.47 | - | 1,041.93 |
| 19 | - | 284.36 | **166.35** | 179.28 | 549.76 | 446.55 |
| 20 | - | 64.16 | **51.23** | 77.72 | 901.22 | 51.33 |
| 21 | - | 273.99 | 384.46 | **101.71** | - | - |
| 22 | - | 69.99 | 325.36 | **38.38** | 331.10 | 83.25 |
| 23 | - | 219.83 | 148.88 | **105.60** | - | 921.71 |
| 24 | - | **66.71** | - | 104.04 | 1,166.18 | 728.24 |
| 25 | 231.02 | 82.18 | 75.33 | **74.93** | 278.38 | 157.90 |
| 26 | **3.20** | 183.71 | 319.49 | 119.51 | 338.69 | 225.64 |
| 27 | 26.03 | **20.94** | 31.05 | 26.29 | 31.74 | 28.56 |
| 28 | - | 104.69 | 75.16 | **73.88** | 1,093.91 | 246.79 |
| 29 | 16.22 | 24.03 | 6.18 | **5.83** | 25.89 | 31.84 |
| 30 | 623.01 | **24.35** | 40.71 | 39.76 | 115.49 | 70.90 |
| 31 | - | 144.15 | 183.09 | **60.75** | 205.21 | 624.58 |
| 32 | - | 221.47 | 74.58 | **68.91** | 453.48 | 194.93 |
| 33 | - | 582.08 | 477.68 | **142.69** | 377.24 | 301.71 |
| 34 | 960.28 | 91.37 | 172.93 | **42.44** | 186.74 | - |
| 35 | 1,198.23 | 104.04 | 121.52 | **85.38** | 1,052.79 | 203.96 |
| 36 | - | 151.65 | 184.78 | **128.23** | - | 545.87 |
| 37 | - | 434.39 | 185.82 | 39.34 | **38.79** | 57.92 |
| 38 | - | 183.02 | **73.48** | 88.23 | 148.20 | 93.28 |
| 39 | 86.48 | **50.87** | 314.47 | 232.05 | 440.41 | 61.29 |

| ID | No Network | $\alpha0$ | OR: White | | OR: Black | |
|---|---|---|---|---|---|---|
| | | | PCN-$b_{max}$ | PCN-$b_{heur}$ | PCN-$b_{max}$ | PCN-$b_{heur}$ |
| 40 | 184.11 | **14.31** | 66.06 | 58.98 | 115.94 | 111.50 |
| 41 | 252.70 | 99.94 | 85.21 | **53.00** | 250.28 | 88.53 |
| 42 | 838.71 | 263.73 | **252.75** | 740.79 | 1,770.01 | 326.05 |
| 43 | - | 474.60 | **101.17** | 104.18 | 150.20 | 366.13 |
| 44 | 345.29 | **57.68** | 127.94 | 96.94 | 103.44 | 125.41 |
| 45 | - | 48.26 | **26.04** | 57.68 | 295.36 | 1,129.93 |
| 46 | - | **432.77** | 491.31 | 465.43 | 528.22 | 499.04 |
| 47 | - | 185.87 | 164.82 | 302.55 | **99.48** | 618.58 |
| 48 | 202.32 | 72.18 | 186.28 | **63.25** | 565.44 | 120.45 |
| 49 | 206.34 | - | 68.18 | **36.49** | 605.91 | 384.63 |
| 50 | - | 131.79 | **56.36** | 56.53 | 161.66 | 223.69 |
| 51 | - | 241.02 | 84.23 | **77.15** | - | 1,224.39 |
| 52 | - | 73.73 | **30.88** | 624.36 | 976.04 | - |
| 53 | 233.12 | 1,153.92 | 50.00 | **41.29** | - | 150.20 |
| 54 | 430.20 | 106.41 | **26.53** | 38.03 | 134.80 | 214.86 |
| 55 | 844.70 | 92.66 | 49.99 | **46.33** | 65.48 | 75.00 |
| 56 | - | 69.00 | **56.28** | 224.92 | 79.94 | 214.85 |
| 57 | - | 350.69 | **281.45** | 585.07 | 1,392.13 | 584.82 |
| 58 | - | **54.89** | 77.31 | 59.04 | 128.94 | 172.72 |
| 59 | - | - | **36.58** | - | - | 266.44 |
| 60 | - | 96.72 | 91.44 | 140.39 | 370.75 | **89.89** |
| 61 | 286.99 | 208.00 | 951.63 | 519.76 | **44.48** | 1,043.74 |
| 62 | 166.89 | 105.47 | 75.49 | **54.51** | 84.17 | 66.29 |
| 63 | - | 178.92 | **50.21** | 81.57 | 170.67 | 94.54 |
| 64 | 1,079.18 | **108.55** | 212.10 | 194.94 | 218.66 | 178.36 |
| 65 | - | 270.09 | **78.17** | 84.74 | 78.95 | 281.10 |
| 66 | - | 126.95 | 55.17 | 165.01 | **51.84** | 71.56 |
| 67 | - | 230.61 | 116.47 | **93.38** | 956.85 | - |
| 68 | - | 706.19 | 123.35 | 230.51 | **118.42** | 140.59 |
| 69 | 264.20 | **45.12** | 382.15 | 296.52 | 1,568.31 | 785.38 |
| 70 | - | 214.32 | **141.80** | 220.62 | - | 729.76 |
| 71 | - | 401.63 | 336.30 | **129.38** | 314.37 | 955.69 |
| 72 | - | 157.31 | **63.33** | 481.37 | - | - |
| 73 | - | 309.80 | **104.24** | 109.34 | 241.93 | 1,449.46 |
| 74 | - | 401.79 | 494.81 | **105.64** | 348.10 | 680.13 |
| 75 | 368.85 | 46.48 | 170.07 | **25.79** | 57.99 | 101.98 |
| 76 | - | 239.43 | **153.76** | 217.91 | 526.48 | 632.97 |
| 77 | 349.00 | 81.59 | 372.46 | 286.34 | **54.16** | - |
| 78 | - | 120.78 | 916.71 | 1,374.30 | **105.49** | 214.10 |
| 79 | **212.18** | 1,161.57 | 631.20 | 1,695.10 | - | - |
| 80 | 377.69 | **63.16** | 189.77 | 140.90 | 725.64 | 437.09 |
| 81 | - | 236.54 | 268.44 | **113.45** | 276.19 | 125.98 |

Table 11: The number of seconds for solving 9x9 Killall-Go problems by FDFPN solver within 30 minutes.

