# OpenReview forum: "AlphaZero-based Proof Cost Network to Aid Game Solving"
_ICLR.cc/2022/Conference — ICLR 2022 Poster_

### Official Review · Reviewer_8SpM · 2021-10-25

**Correctness:** 4
**Technical Novelty And Significance:** 2
**Empirical Novelty And Significance:** 2
**Recommendation:** 5
**Confidence:** 3

**Main Review:**

This paper makes AlphaZero becomes faster by modifying the training target of the AlphaZero algorithm, such that it prioritizes solving the game quickly, rather than winning, and training knowledge-based networks. However, I have several concerns:

1. I think the discussion about the relationship between playing games and solving games is questionable. In this paper, they argue, “There are two main goals in the pursuit of strong game-playing agents. The first is to push the boundaries of artificial intelligence since games can be seen as simplified models of the real world. The second involves finding game-theoretic values, or outcomes given optimal play, for various games (van den Herik et al., 2002). These two closely related yet separate goals are commonly referred to as playing and solving games, respectively.” In other words, authors think playing games and solving games are two different goals. In my opinion, solving games is the first step for playing games, i.e., playing games is the goal for solving games. Otherwise, what is the goal for (after) solving games? Our goal should be using the computed strategy in a game, not just solving a game.

2. It is straightforward that this paper makes AlphaZero faster by modifying the target and using existing heuristic knowledge. As we know that, AlphaZero achieved super-human playing levels for many games without hand-crafted expert knowledge. We can expect that AlphaZero can perform better by adding expert knowledge. In addition, this work just incrementally modifies/exploits existing methods/heuristic knowledge.

3. The challenge for this problem in this paper is unclear. We know that the challenge for game solving/playing is the difficulty of searching the huge action space. If the problem in this has the same challenge, how does the proposed method mitigate this difficulty (in contrast to AlplaZero)?

4. Experiments can be improved. Current experiments are only on 15x15 Gomoku and 9x9 Killall-Go problems. I suggest that the authors use the games in the original AlphaZero paper, e.g., the original Go and Chess, to better evaluate the proposed approach. AlphaZero can solve the original Go and Chess with the huge action space, and then we need to know if the proposed method can better solve them or not.


**Summary Of The Paper:**

As we know that, from AlphaGo to AlphaZero, less and less expert knowledge is used. In this paper, however, the authors make AlphaZero becomes faster by modifying the training target of the AlphaZero algorithm, such that it prioritizes solving the game quickly, rather than winning, and training knowledge-based networks. Experimental evaluation is provided on 15x15 Gomoku and 9x9 Killall-Go problems.

**Summary Of The Review:**

It is straightforward that this paper makes AlphaZero faster by modifying the target and using existing heuristic knowledge.

---

> ### Author Response · Authors · 2021-11-14
> **Response to Reviewer 8SpM**
>
> Thank you for your valuable comments. We answer your concerns as follows.
>
> * For concern 1, please see the general response. We want to clarify that we do not claim to make AlphaZero faster, nor is PCN meant for game playing. We hope that the citations that we include in the general response indicate that the distinction between playing and solving are widely accepted. If our explanations are still lacking, we would love to further clarify any misunderstandings on this point.
> * For concern 2: In terms of additional expert knowledge, we compared between NDK and 4T for 15x15 Gomoku. The results demonstrate two points: (1) With the help of 4T, the solver is able to solve more problems than NDK. (2) PCN is able to solve more problems than the original AlphaZero in either case.
> The first corroborates your expectations in that the solver can potentially perform better with more expert knowledge. We feel, however, that our main contribution is the second point. Namely, regardless of whether we use these heuristics, PCN is always the better choice when solving games.
> * For concern 3, please see the general response.
> * For concern 4: As we stated earlier in this response, playing strength using PCN is beside the point. We do concede that it is worth trying out, so we included an experiment comparing the playing strength between AlphaZero and PCN in Appendix A. The results show that for Go, PCN is stronger than the original AlphaZero.
> As for the choice of benchmarks, there are a variety of papers that improve upon the AlphaZero idea, which do not use the same set of benchmarks as the original AlphaZero paper [1-5].
>
> [1] Gao, Chao, et al. "Three-Head Neural Network Architecture for AlphaZero Learning." (2019).
>
> [2] Cazenave, Tristan, et al. "Polygames: Improved zero learning." ICGA Journal Preprint (2020): 1-13.
>
> [3] Li, Xiali, et al. "A Game Model for Gomoku Based on Deep Learning and Monte Carlo Tree Search." Chinese Intelligent Automation Conference. Springer, Singapore, 2019.
>
> [4] Wang, Hui, et al. "Hyper-parameter sweep on alphazero general." arXiv preprint arXiv:1903.08129 (2019).
>
> [5] Soemers, Dennis JNJ, et al. "Deep learning for general game playing with ludii and polygames." arXiv preprint arXiv:2101.09562 (2021).

---

> > ### Comment · Reviewer_8SpM · 2021-11-18
> > **Response to authors**
> >
> > Thanks for your clarification. I still have concerns on the contribution. If solving the game is for exact solution value, then what is the solution concept here? Does the proposed algorithm theoretically converge to that exact solution?
> >
> > In addition, I do not think the challenge has been stated in the general response.
> >
> > By the way, it is a pity that the above references [1-5] mentioned in the response, for not using the same set of benchmarks as the original AlphaZero paper, were not published on the top-tier conference/journal.

---

> > > ### Author Response · Authors · 2021-11-20
> > > **Response to Reviewer 8SpM**
> > >
> > >
> > > > If solving the game is for exact solution value, then what is the solution concept here?
> > >
> > > - The exact value refers to the optimal outcome for both players, e.g., draw for Checkers [1], first player win for Connect4 (and those in [2]). The solution takes the form of a strategy; for two-player games, the strategy is a so-called solution tree [4, 5]. Let's use Chess as an example for illustration. Even though the solution for chess itself is not yet known, plenty of endgames have been solved [3]. E.g. if a specific endgame is proven for Black, the exact value is a win for Black, and the solution tree contains all optimal responses to all White moves, to reach a winning outcome for Black.
> > >
> > > > Does the proposed algorithm theoretically converge to that exact solution?
> > >
> > > - Our proposed algorithm only learns a *heuristic* that estimates the size of the proof tree. This heuristic can be used to guide any complete search algorithm. By definition, a complete search algorithm will find a solution (a solution tree) if a solution exists; our heuristic speeds this process up significantly. When the search finds the solution, we are guaranteed to obtain the game-theoretic value, so in that respect the combined algorithm always converges, given sufficient time and computational resources.
> > >
> > > > In addition, I do not think the challenge has been stated in the general response.
> > >
> > > - Could you elaborate on what kind of response you would like to see? In terms of why finding a solution is challenging, in the general response we state the following explicitly:
> > > 	- Solving is harder than playing.
> > > 	- In stark contrast to AlphaZero's superhuman strength in playing 19x19 Go, only 5x6 Go has been proven to date.
> > > 	- Beyond Go, there are trends that show exact solutions for puzzles or non-game applications (e.g. chemistry) are also highly worthy of investigation.
> > > - In the general response, we also cite papers [1, 6-9] that look for exact solutions in many notable conferences (NeurIPS, ICLR) and journals (Science, Nature).
> > >
> > > > About the references we list in the last reply.
> > >
> > > - We think where those references were published is beside the point. We included those references simply to illustrate that AlphaZero is a very general algorithm that can be used across a wide range of benchmarks, and that our heuristic, being a game-independent method, can also be used across many different games and settings.
> > >
> > > [1] Jonathan Schaeffer, Neil Burch, Yngvi Bjornsson, Akihiro Kishimoto, Martin M ¨ uller, Robert Lake, ¨ Paul Lu, and Steve Sutphen. Checkers is solved. Science, 317(5844):1518–1522, 2007.
> > >
> > > [2] H Jaap van den Herik, Jos W H M Uiterwijk, and Jack Van Rijswijck. Games solved: Now and in the future. Artificial Intelligence, 134(1-2):277–311, 2002.
> > >
> > > [3] Syzygy endgame tablebases, https://syzygy-tables.info/
> > >
> > > [4] Wim Pijls and Arie de Bruin. Game tree algorithms and solution trees. Theoretical computer science, 252(1-2):197–215, 2001.
> > >
> > > [5] George C. Stockman. A minimax algorithm better than alpha-beta? Artificial Intelligence, 12(2): 179–196, 1979.
> > >
> > > [6] Stephen McAleer, Forest Agostinelli, Alexander Shmakov, and Pierre Baldi. Solving the Rubik’s cube with approximate policy iteration. In International Conference on Learning Representations, 2018.
> > >
> > > [7] Forest Agostinelli, Stephen McAleer, Alexander Shmakov, and Pierre Baldi. Solving the Rubik’s cube with deep reinforcement learning and search. Nature Machine Intelligence, 1(8):356–363, 2019.
> > >
> > > [8] A. Kishimoto, Beat Buesser, B. Chen, and A. Botea. Depth-first proof-number search with heuristic edge cost and application to chemical synthesis planning. In Advances in Neural Information Processing Systems, 2019.
> > >
> > > [9] Marwin HS Segler, Mike Preuss, and Mark P Waller. Planning chemical syntheses with deep neural networks and symbolic ai. Nature, 555(7698):604–610, 2018.

---

> > > > ### Comment · Reviewer_8SpM · 2021-11-22
> > > > **RE**
> > > >
> > > > Thanks for your response.
> > > >
> > > > I suggest the authors clearly state the solution concept by explaining what properties an optimal strategy should have.
> > > >
> > > > About challenge, the authors should discuss why solving is hard (should not just mention that it is harder than playing) and explain how the proposed algorithm mitigates this difficulty.
> > > >
> > > > I cannot see a good reason for not using the same set of benchmarks as the original AlphaZero paper.
> > > > In the worst case, the proposed algorithm still needs the complete search algorithm. Therefore, I am wondering if the proposed approach works on larger game, e.g., 19x19 Go.

---

> > > > > ### Comment · Reviewer_b9Ze · 2021-11-22
> > > > > **Solving, compared to play**
> > > > >
> > > > > I'd describe the difference as follows. Solving is finding a proof of the maximum obtainable value, against any possible opponent policy (including the optimal opponent play against our policy). This is an absolute statement: a true maximum. Play is trying to find a series of good moves, along a trajectory in a game episode. This is a fuzzy statement: the objective is also trying to maximise the value, but there is neither a guarantee or an expectation that it'll happen.
> > > > >
> > > > > Is the concern that it's not clear why finding a proof is harder than picking a good action, so that something like the description above is also insufficient?
> > > > > Or is the concern that this kind of description is missing? I read footnote 1 as a different, shorter phrasing of the first part of the description above, assuming that play is well-understood. It does at least describe the properties of an optimal strategy.
> > > > >
> > > > > Solving 19x19 Go (or even chess) with a general algorithm is not really a pratical question: the answer -- a proof -- is of size ~sqrt(game size).

---

> > > > > ### Author Response · Authors · 2021-11-22
> > > > > **More detailed official response incoming**
> > > > >
> > > > > We'd like to thank reviewer b9Ze for joining the discussion. They have more or less answered the questions as we would have, but from a fresh perspective. We will also add an official response at a later hour after conferring with other co-authors in a different time zone.

---

> > > > > ### Author Response · Authors · 2021-11-23
> > > > > **Response to Reviewer 8SpM**
> > > > >
> > > > > > I suggest the authors clearly state the solution concept by explaining what properties an optimal strategy should have.
> > > > > - We have already added the definition for a game's solution in footnote 1 as reviewer EgiY suggested. Section 2.1 illustrates more details about the properties of an optimal strategy, including the solution tree.
> > > > >
> > > > > > About challenge, the authors should discuss why solving is hard (should not just mention that it is harder than playing) and explain how the proposed algorithm mitigates this difficulty.
> > > > > - In addition to stating solving is harder than playing, we also list many papers that show solving is a challenging problem, and that it can be used in other fields (e.g. chemistry) in the previous responses.
> > > > > - As for our method as an improvement to existing methods, in the third paragraph of the introduction, we mention: "However, a limitation of using these techniques is that the networks trained with the objective of strong play may not be well-suited for obtaining game-theoretic values. ... This paper presents a novel approach to solving these problems, while still tending to choose moves that lead to the fastest solutions."
> > > > >
> > > > > > I cannot see a good reason for not using the same set of benchmarks as the original AlphaZero paper. In the worst case, the proposed algorithm still needs the complete search algorithm. Therefore, I am wondering if the proposed approach works on larger game, e.g., 19x19 Go.
> > > > > - As reviewer b9Ze mentioned, solving 19x19 Go is currently not a practical question. To expand on their comments, by "game size", they are likely referring to the game tree size, which some have estimated to be 10^360 [1, 2], i.e. the proof is in the order of sqrt(10^360). No researcher has been able to prove 6x6 Go, let alone 19x19.
> > > > >
> > > > > [1] H Jaap van den Herik, Jos W H M Uiterwijk, and Jack Van Rijswijck. Games solved: Now and in the future. Artificial Intelligence, 134(1-2):277–311, 2002.
> > > > >
> > > > > [2] Allis, Louis Victor. Searching for solutions in games and artificial intelligence. Wageningen: Ponsen & Looijen, 1994.

---

### Official Review · Reviewer_b9Ze · 2021-10-28

**Correctness:** 3
**Technical Novelty And Significance:** 3
**Empirical Novelty And Significance:** 3
**Recommendation:** 8
**Confidence:** 4

**Main Review:**

In general, I like the ideas and results in this work, and would like to see it published. My main concern is that the paper has a organisational issue, with a recurring pattern of discussing things before they are introduced. I like the author's discussion of the background material, but found the discussion of the algorithm hard to follow, despite being familiar with both PNS and AlphaZero. Subjectively, it felt like the extra details I wanted to know were not where I expected them to be.

One suggestion for an easy-to-follow structure would be reducing the current level of detail in section 3.1, adding a brief mention of a similar notion of minimum disproof tree size, and adding a couple of sentences foreshadowing how approximations of these tree sizes will be the heuristic generated by the selfplay loop and then used in a modified PN solving algorithm. Then move the both the selfplay and solving algorithms to the beginning of section 3.2, and subsequently discuss the correctness / design choices afterwards.

--- Specific questions and comments ---
There is an unadressed issue of total computation time. While the heuristic generated during training is general, that training time is amortised over the number of problems that actually get solved. Where is that break-even point of 1500h training vs 30m search? For example, presumably for one problem the 1500h is best spent on a no-heuristic version of that problem, and for millions of problems that 1500h training time is cheap. 1000s of problems?  The 90 problems tested?

"Then the inductive definition of n(s) is n(s) = 1 + sum_i=1 n(s_i)."
The definition must include a distinction between OR and AND node, or the player to act, or ...? Otherwise for state s leading to a loss and a win, n(s) = 1 + n(L) + n(W) = 1 + 1 + inf = inf. The following discussion corrects this, but the statement as given here does not seem to be technically correct.

For clarity, consider indexing the proof tree size by the player of interest, as in n_p(s). This would help with two things. First, it would then be clear that n_p(t) is the size of the minimum proof tree of t for some particular player of interest, not the current player -- it is possible to misinterpret "number of tree nodes, denoted by n(s), for the player at s to win". Second, it would help smooth the jump from the fixed root of the selfplay training, to (dis)proving wins at other states s -- this works as long as the player of interest P(s) matches the P(s_0) for the initial game state s_0 used in the selfplay training, which is why OR/AND player is fixed.

"In practice, with an imperfect heuristic, the order in which children of AND nodes are searched will still have implications on the solution size, regardless of which search algorithm is used (MCTS or PNS)."
Is it possible to clarify why directly in the text, so the reader isn't left waiting to discover why, or stuck thinking about why?  Something like
"with an imperfect heuristic, the order in which children of AND nodes are searched matters because one child might lead us to discovering the mis-ordering faster than other children."

"Instead, we take advantage of the self-play phase in AlphaZero to collect game episodes"
This would be a good place to note that it's selfplay using a modified MCTS search -- don't leave this for the reader to eventually discover later on.

"There are some similarities and differences between our heuristic and the concept of PNs in PNS. First, PN/DN are meant to change as PNS progresses and new nodes are expanded and evaluated. In that sense, the PN/DN are a dynamic quantity used to signify the least number of nodes that need to be expanded to prove/disprove each node at that point in the search. In contrast, n̄(s) is a static estimate of an oracle number that represents the minimum proof tree size."
This starts with similarites and differences. What are the similarities?
Would it be simpler (and still correct) to describe PNs as a lower bound on n(s) - current_tree_size(s), the remaining work to build a proof tree, using an uninformed estimate of n(s)~=n̄(s)=1 for all unexpanded nodes?

"Second, PNS makes no assumptions ...  m̄(s)"
The introduction of m̄() at this point feels like a forward reference.  n̄() is related to the preceding discussion of minimum proof tree size n(), but this is missing for m̄. Consider adding a sentence at end of the discussion of minimum proof tree size, noting there is a similarly defined minimum disproof tree size m(s) that shows s is not a win.

"In our method, while OR nodes behave similarly in that the smallest n̄ value is chosen ... when training the heuristic network."
I found this paragraph confusing -- it is another forward reference to an algorithm which has not yet been given. Either the algorithm needs to precede this discussion, or this paragraph needs to be re-written to be a gentle, more general discussion of what properties a game-solving algorithm using n() and m() should have.

Experimental section:
Consider combining table 1 and 2. Comparing MCTS and PN seems like a natural thing to do, and splitting into two tables makes this harder.  If the authors prefer splitting the table so it is smaller, consider splitting by game instead?

Experimental section:
The choices in b_heur seem fairly safe, at least with respect to ratios of n(s) and n(s_child). Particularly using a b factor of the number of legal moves, versus all possible moves on an empty board. Given that, it seems like a surprise that b_max is often better than b_heur, and worth mentioning. Do the authors have a (short) explanation or conjecture that can be added? Is this not actually a surprise?


**Summary Of The Paper:**

This paper uses a modified AlphaZero MCTS training loop to generate proof tree size heuristics, for use in MCTS or proof number search (PNS). The authors demonstrate that both heuristics outperform no heuristic and a existing heuristic method based off of standard search policy and value functions. The ideas and results are a nice step forward in a search domain that is is not always well-addressed by classical value-maximising search.

**Summary Of The Review:**

I am generally positive about this paper. There are a few small issues to correct, and some things which could be expanded, but the basic pieces of a solid submission are there. Howeve, I do think that the readability of the paper could be substantially improved through some reorganisation and rewriting of some sections.

---  Thoughts after revisions and discussion ---
I think the readability of the paper could has improved. Even if it could be further improved, I think the paper is in a reasonable state for publication.
------

---

> ### Author Response · Authors · 2021-11-14
> **Response to Reviewer b9Ze**
>
> Thank you for your valuable comments. We answer your concerns as follows.
>
> * Thank you for your suggestions, we will try to reorganize the structure of Sections 3.1 and 3.2 in the next revision.
> * For question 1 (about computation time): This is a very interesting question. In terms of wall-clock time, we believe that the process of solving the games will be much longer than the training process. For example, we need 1500 GPU hours to train a Gomoku agent, which can then be used for anything up to and including Gomoku from the empty board. On the other hand, the problems we picked for the benchmark are all much smaller than the empty board, which is why the search time seemed to be measured in the neighborhood of 30 min. Also, in practice, the most expensive part of the training process is the self-play, which can be parallelized almost trivially.
> * For questions 2 & 3 (about n(s)): In this paper, n(s) denotes the number of tree nodes for the player of interest to win. We realize that the current way it is written is ambiguous and prone to misunderstanding, so we will correct this. Thank you for catching this for us!
> * For questions 4-8, we will revise them directly in the next revision.
> * For the first point on experiments: We agree that it is almost a reflex to want to compare MCTS and PNS. However, our focus is on the comparison between different networks (AlphaZero and PCN) as heuristics, which is search-independent. To put it bluntly, we organized the tables this way in order to discourage readers from comparing search algorithms since we are not making any claims as to which search algorithm fits better with AlphaZero and PCN for solving games. However, if this explanation does not sufficiently satisfy your concerns, we can revise it as you have suggested.
> * For the second point on experiments: Preliminary experiments show that when more complex heuristics are added, b_heur is on par or even better than b_max. We think the limited scope of heuristics used may have led to inaccuracies, which then impacted the overall search efficiency. Since the main point of PCN is that it is general and does not necessarily require expert knowledge, we omitted complicated heuristics to illustrate the main point more succinctly.

---

> > ### Comment · Reviewer_b9Ze · 2021-11-15
> > **Response^2**
> >
> > - I'm satisfied with an intentional choice to not encourage directly comparing MCTS vs. PNS.
> > - The explanation about about b_max and b_heur sounds good to me, especially given the other experimental results. Adding some variant of those two sentences to the paper would likely answer similar question from future readers.
> >
> > I remain positive about the contributions in this paper, and hope that some changes for clarity help convince the other reviewers.

---

> > > ### Author Response · Authors · 2021-11-17
> > > **Response to Reviewer b9Ze**
> > >
> > > We have revised 3.1 and 3.2 to better illustrate a roadmap of how the subsequent subsections are organized. We generally agree with your comments on how to present the core ideas in a clearer way, but are hesitant to make massive changes to the organization of the paper, given how the current reviewer comments are for the previous version of the paper. If the current revision is still hard to follow, please let us know and we will do our utmost to improve it.

---

> > > > ### Comment · Reviewer_b9Ze · 2021-11-17
> > > > **Comment after revision**
> > > >
> > > > The changes made do correct many of the out-of-order references, and it does help a subsequent read-through. I do still think that some more substantial re-ordering could be made to further improve readability, but understand a reluctance to make major changes based on a single reviewer's suggestion.

---

> > > > > ### Author Response · Authors · 2021-11-20
> > > > > **Response to Reviewer b9Ze**
> > > > >
> > > > > Thank you for your consideration and for raising your score!
> > > > > We really appreciate the effort you put into your reviews and giving valuable suggestions to us. Thank you again.

---

### Official Review · Reviewer_CHG3 · 2021-10-31

**Correctness:** 3
**Technical Novelty And Significance:** 3
**Empirical Novelty And Significance:** 3
**Recommendation:** 8
**Confidence:** 3

**Main Review:**

**Primary Strengths**:
1) Well-written paper, almost everything is clear.
2) Interesting contribution.
3) Good empirical results.

**Primary Weaknesses**:
I really don't have much to remark here. I do have several minor issues (like some notation) and a few small points that confused me, but I expect these should be relatively easy to resolve; see detailed comments below.

---

**Detailed Comments**:
- Acronym FDFPN used in abstract without fully writing it out, and (arguably unlike MCTS) this one really isn't common enough to assume that every reader will know what it is.
- When putting words/phrases in quotes, use backticks instead of '' on the left-hand side
- In Eq. (1), the symbol on the left-hand side of the equality (currently $a$) should probably be different from the symbol used under the argmax (also $a$) on the right-hand side. I'd suggest using $a^*$ on the left-hand side.
- First paragraph of 3.1 ends with "Then the inductive definition of $n(s)$ is $n(s) = 1 + \sum_{i=1}^b n(s_i)$." But the next paragraph immediately seems to contradict this, because it's actually only correct for AND nodes, not for OR nodes.
- "Then, $n(s_{AND}) = 1 + \sum_{s_i} n(s_i)$ for all children $s_i$" --> the "for all children $s_i$" phrasing is a bit confusing since the sum already implies a "for-all loop". Should probably be changed to something like ", where $s_i$ are the children of $s_{AND}$."
- In Subsection 3.2, I got really curious about whether there is any particular reason for not adding $+ 1$ to the definitions of the $\bar{n}(s)$ heuristics, as you also would for the real proof/disproof numbers? This would be worth clarifying.
- Final sentence of first paragraph of 4.2: the results really don't back up this claim about "closing the gap in performance" in my opinion. The gap is almost identical.

---

**After authors' response**: I am satisfied with the authors' response and revisions.

**Summary Of The Paper:**

This paper describes an AlphaZero-like approach to train networks from self-play, where the networks are trained to predict (logarithms of) proof costs / disproof costs, which are heuristics closely related to proof numbers / disproof numbers. Empirical evaluations show that these heuristics can be used by MCTS-based solvers as well as solvers based on Proof Number Search, and effectively enable both types of solvers to solve problems more efficiently / solve more problems in fixed time budgets.

**Summary Of The Review:**

A well-written paper, with no major issues. Some minor issues but I expect these should be relatively easy to clear up or resolve.

---

> ### Author Response · Authors · 2021-11-14
> **Response to Reviewer CHG3**
>
> Thank you for your valuable comments. We answer your concerns as follows.
>
> * For comment 6 (about not adding +1): The reason why we did not add 1 for $\bar{n}(s)$ is because the values are meant to be exponential in scale, since we are estimating the size of the proof tree. In the end the log value is used, for which the 1 is simply inconsequential. For clarity and consistency however, we should explain more explicitly in the revision.
> * For comment 7 (about the gap): We agree that an improvement from 5 to 4 is not obvious in Table 1. We included this description because we had observed this trend during preliminary experiments. However, now that you point out the formal results do not support this sufficiently, it is glaringly obvious that unless more experiments are conducted, we cannot simply claim that this trend is significant. We will remove this part from the experiments section and mention it as worthy of future work.
> * We will revise accordingly for your other comments in the next revision.

---

> > ### Comment · Reviewer_CHG3 · 2021-11-18
> > **Response to Authors**
> >
> > Thanks for your clarifications, and I also just noticed a new version of the PDF has been uploaded with various comments addressed. I have updated my score accordingly.

---

> > > ### Author Response · Authors · 2021-11-20
> > > **Response to Reviewer CHG3**
> > >
> > > Thank you for raising the score! We are really grateful for your valuable comments and suggestions.

---

### Official Review · Reviewer_EgiY · 2021-11-01

**Correctness:** 4
**Technical Novelty And Significance:** 2
**Empirical Novelty And Significance:** 2
**Recommendation:** 5
**Confidence:** 3

**Main Review:**

Strength:
The proposed PCN network equipped with AlphaZero (MCTS in precise) or FDFPN solver is able to solve the game more efficiently then the original algorithm. Sufficient experiment results and illustrations are provided.

Weaknes:
1. The term, solving a game, is ambiguous without a formal description in math. It is easily confused with computing a Nash equilibrium or other solution concept for a game. This brings difficulty to understand what problem this paper is going to solve.
2. The contribution of this work is not well discussed in this paper. I wonder whether it is significant to quickly solve a game from scratch. It seems that a pretrained AlphaZero or other programs is als oable to solve the game (as what they didi in generating the game examples). Moreover, even the importance of solving a game is not well described in this paper.

**Summary Of The Paper:**

This paper focus on solving a game, i.e. deciding the win/lose outcome for each game state. In order to finish the task in limited time, it is necessary to expand as little leaf nodes as possible when searching on the game state tree. The number of such leaf nodes is defined as proof cost and the paper proposes to set it as the new learning target based on the AlphaZero learning framework. Experiments are conduct to verify the ability of the model by solving $15 \times 15$ Gomoku and $9 \times 9$ Killall-Go games in limited time.

**Summary Of The Review:**

This paper proposed a new learning objective that enhance AlphaZero and FDFPN to solve a game. However, without a sufficient discussion about the significance of the studied problem, I do not recommend acceptance.

---

> ### Author Response · Authors · 2021-11-14
> **Response to Reviewer EgiY**
>
> Thank you for your valuable comments. We answer your concerns as follows.
>
> * For weakness 1: Our definition follows that of [1]. We understand that there is an argument for a more explicit definition within the main text for self-containedness, so if you think this is necessary, we can include the definition from [1].
> * We hope the general response regarding playing versus solving addresses your other concerns.
>
> [1] H Jaap van den Herik, Jos W H M Uiterwijk, and Jack Van Rijswijck. Games solved: Now and in the future. Artificial Intelligence, 134(1-2):277–311, 2002.

---

> > ### Comment · Reviewer_EgiY · 2021-11-15
> > **Response to Author's response**
> >
> > I can see the difference between playing a game and solving a game. Thanks for making it more clear in general response. However, it is still necessary to include the significance of solving games in this work. Similarly, the definition should be included for self-containedness.
> >
> > Despite the fact that these two issues can be added in the revision, I still have concern on the contribution of this work. It seems that (reminder me if I am wrong) the proposed algorithm can only solve the games, which are already known can be solved, with a much efficient way.

---

> > > ### Author Response · Authors · 2021-11-17
> > > **Response to Reviewer EgiY**
> > >
> > > * With consideration to the page limits, the significance of game solving is currently covered in the second and third paragraphs in the Introduction.
> > > * We use 15x15 Gomoku and 9x9 Killall-Go as benchmarks in this paper for the following reasons. The former has been solved as Black win, however, the latter has not yet been solved. We chose Gomoku since it is more friendly to readers who are not familiar with the rules of Go. With Gomoku's threat property, it is also highly suitable to illustrate the difference between b_heur and b_max, e.g. as shown in Fig. 3. We chose 9x9 Killall-Go since Go is a very representative benchmark in the board game field due to its difficulty in designing accurate evaluation functions. This is one of the major reasons why DeepMind chose Go to demonstrate AlphaGo/AlphaGo Zero. As a side note, Killall-Go is similar to so-called life-and-death problems (or called tsumego) in Go, which can be seen as a preliminary step in solving Go. Although AlphaZero has achieved superhuman levels in playing 19x19 Go, only 5x6 Go has been proven to date, which is still far away from 19x19. We hope that with our method and previous Go heuristics, it will be possible to extend the solvable size of Go.

---

> > > > ### Comment · Reviewer_EgiY · 2021-11-18
> > > > **Thanks for your clarification.**
> > > >
> > > > Thanks for your clarification.

---

> > > > > ### Author Response · Authors · 2021-11-22
> > > > > **Addressing any remaining questions**
> > > > >
> > > > > Thank you for taking the time to review and provide constructive comments on our paper. Have we addressed your concerns sufficiently in the responses and revision? We would love to take advantage of the remaining time to address any further questions you might have.

---

### Author Response · Authors · 2021-11-14
**General response to all Reviewers**

Thank you for your thoughtful reviews and constructive comments! We are currently taking your comments into consideration for the next revision of this paper, which we will upload soon.

Before addressing the individual questions and concerns, we would like to first clarify the difference between playing and solving games again:
1. "Playing" generally refers to a non-perfect, best-effort attempt to achieve a game's goals. In contrast, "solving" refers to the process of finding the "game-theoretic" [1] outcome for a game. We know that 15x15 Gomoku has already been solved as a first player win. In contrast, despite how well AlphaZero plays Go and chess, these two games have not yet been solved. If AlphaZero had solved Go or chess, it would no longer be possible to create a stronger program than AlphaZero in these two games. This is simply not the case.
2. Therefore by definition, solving is harder than playing. Case in point, superhuman level checkers programs existed for a long time [2] before the game of checkers was definitively solved [3]. As a side note, solving checkers was an achievement that was widely recognized as significant, warranting a publication in the esteemed journal Science.
To illustrate, we quote a relevant passage in [3]: "*In 1963, [a machine learning] program played a match against a capable player, winning a single game. This result was heralded as a triumph for the fledgling field of AI. Over time, the result was exaggerated, resulting in claims that checkers was now "solved".*" The milestone of a computer program beating a human player in 1963, in contrast with the game being solved in 2007, shows how wide the gap of difficulty is between playing relatively well and solving the game outright. Go is currently in the same situation. In stark contrast to AlphaZero's superhuman strength in playing 19x19 Go, only 5x6 Go has been proven to date [4].
3. There is a fundamental difference in playing versus solving; as respectable and exciting as achievements in the former category are, they are ultimately heuristics, as opposed to perfection and exactness for the latter category. Beyond Go, there are trends that show exact solutions for puzzles [5, 6] or non-game applications [7, 8] (e.g. chemistry) are also highly worthy of investigation. AlphaZero is a strong tool that we can leverage to aid in finding exact solutions because it does not require human knowledge. We simply wish to share with this paper that there are effective, non-trivial methods to improve AlphaZero as a tool for finding exact solutions.

[1] H Jaap van den Herik, Jos W H M Uiterwijk, and Jack Van Rijswijck. Games solved: Now and in the future. Artificial Intelligence, 134(1-2):277–311, 2002.

[2] J. Schaeffer, One Jump Ahead (Springer-Verlag, New York, 1997).

[3] Jonathan Schaeffer, Neil Burch, Yngvi Bjornsson, Akihiro Kishimoto, Martin M ¨ uller, Robert Lake, ¨ Paul Lu, and Steve Sutphen. Checkers is solved. Science, 317(5844):1518–1522, 2007.

[4] Erik C D van der Werf and Mark H M Winands. Solving go for rectangular boards. ICGA Journal, 32(2):77–88, 2009.

[5] Stephen McAleer, Forest Agostinelli, Alexander Shmakov, and Pierre Baldi. Solving the Rubik’s cube with approximate policy iteration. In International Conference on Learning Representations, 2018.

[6] Forest Agostinelli, Stephen McAleer, Alexander Shmakov, and Pierre Baldi. Solving the Rubik’s cube with deep reinforcement learning and search. Nature Machine Intelligence, 1(8):356–363, 2019.

[7] A. Kishimoto, Beat Buesser, B. Chen, and A. Botea. Depth-first proof-number search with heuristic edge cost and application to chemical synthesis planning. In Advances in Neural Information Processing Systems, 2019.

[8] Marwin HS Segler, Mike Preuss, and Mark P Waller. Planning chemical syntheses with deep neural networks and symbolic ai. Nature, 555(7698):604–610, 2018.

---

### Author Response · Authors · 2021-11-17
**Summary of Revision**

Dear reviewers,

We have uploaded a new revision of our paper. We list the changes as follows:
* Write the full name of FDFPN in the abstract.
* Add definition for a game's "solution" in Introduction as a footnote.
* Change the format of the quotes.
* Change $a$ to $a^*$ in Eq. (1).
* Change "for all children $s_i$" to ", where $s_i$ are the children $s_{AND}$".
* Added a roadmap for all subsections in section 3 (added to the end of 3.1).
* Add explanation for not adding 1 to the $\bar{n}(s)$ values in Subsection 3.2.
* Add a possible explanation for why b_heur is not consistently better than b_max in the discussion section.

---

### Decision · Program_Chairs · 2022-01-20

**Decision:**

Accept (Poster)

**Comment:**

This paper modifies the AlphaZero algorithm to generate proof tree size heuristics and shows empirical improvements over standard search algorithms. This is an interesting distinction that might lead to algorithms with distinct play styles and a deeper understanding of the games that we apply our agents to.

The two positive reviewers felt that it was a solid contribution, worthy of publication. There were some questions regarding the clarity of the writing that were addressed in the discussion phase. The two reviewers that gave lower scores felt that the paper did not do a sufficient job motivating the work and distinguishing itself from the literature. Ultimately, I agree with the positive reviewers, and it is my opinion that the revised version is acceptable for publication.